# UNComp: Uncertainty-Aware Long-Context Compressor for Efficient Large Language Model Inference

## Abstract

Deploying large language models (LLMs) is challenging due to their high memory and computational demands, especially during long-context inference. While key-value (KV) caching accelerates inference by reusing previously computed keys and values, it also introduces significant memory overhead. Existing KV cache compression methods—such as eviction and merging—typically compress the KV cache after it is generated and overlook the hidden states, failing to improve the speed of the prefilling stage. Additionally, applying a uniform compression rate across different attention heads can harm crucial retrieval heads in needle-in-a-haystack tasks due to excessive compression. In this paper, we propose *UNComp*, an uncertainty-aware compression scheme that leverages matrix entropy to estimate model uncertainty across layers and heads at the token sequence level. By grouping layers and heads based on their uncertainty, *UNComp* adaptively compresses both the hidden states and the KV cache. Our method achieves a 1.6× speedup in the prefilling stage and reduces the KV cache to 4.74% of its original size, resulting in a 6.4× increase in throughput and a 1.4× speedup in inference with only a 1.41% performance loss. Remarkably, in needle-in-a-haystack tasks, *UNComp* outperforms the full-size KV cache even when compressed to 9.38% of its original size. Our approach offers an efficient, training-free Grouped-Query Attention paradigm that can be seamlessly integrated into existing KV cache schemes.

## 1 Introduction

The proliferation of large language models (LLMs) has led to unprecedented advancements in natural language processing (Achiam et al., 2023; Kaplan et al., 2020), enabling capabilities ranging from simple text generation to complex reasoning and dialogue. However, deploying and scaling LLMs are significantly hindered by extensive memory requirements and computational costs (Shazeer et al., 2017), especially during long-context inference. Processing long contexts leads to significant computational time during the prefilling stage, and the "attention sink" phenomenon (Xiao et al., 2023) during decoding impedes efficient long-text generation.

To mitigate these issues, *KV caching* (Pope et al., 2023; Liu et al., 2024b) stores and reuses keys and values to avoid redundant computations, improving inference speed. However, the memory overhead of maintaining the KV cache remains prohibitive for long contexts (Liu et al., 2024b), prompting the development of methods to reduce KV cache size while preserving performance.

Existing optimization methods include: *i*) **Eviction Strategies** (Ge et al., 2023; Zhang et al., 2024d; Li et al., 2024; Zhang et al., 2024c); *ii*) **Merging Strategies** (Liu et al., 2024b; Wan et al., 2024; Wang et al., 2024; Zhang et al.); *iii*) **Quantization** (Hooper et al., 2024; Zhang et al., 2024a; Liu et al., 2024e); *iv*) **Compressing KV Cache Heads** (Ainslie et al., 2023; Shazeer, 2019; Liu et al., 2024a; Yu et al., 2024). However, these methods often compress the KV cache after generation and neglect hidden states, failing to speed up the prefilling stage. Moreover, uniform compression across attention heads can degrade important retrieval heads due to over-compression.

Multi-Query Attention (MQA) (Ainslie et al., 2023; Yu et al., 2024; Brandon et al., 2024) reduces attention heads during inference by grouping heads and using a single head per group, significantly

reducing memory and improving speed while maintaining performance comparable to Multi-Head Attention (MHA). However, these methods typically require fine-tuning or training from scratch, which can be costly. A training-free approach to group heads during inference is more practical.

Inspired by these insights, we propose estimating the internal *uncertainty* across layers and heads within the KV cache and hidden states. Previous metrics based on cumulative attention distributions focus on token-level importance but overlook sequence-level sampling strategies such as estimating heads and layers. By measuring the effective information via *matrix entropy*, we quantify uncertainty across heads and layers, enabling adaptive compression of the KV cache and pruning of hidden states for faster inference. Our key contributions are:

1. We propose a novel method based on matrix entropy to explore the uncertainty within the hidden states and KV cache, analyzing their information compression patterns. This approach enables grouped compression of heads and layers. By compressing the hidden states, we achieve a 1.6× speedup in the prefilling stage in a single batch.

2. We employ a training-free approach to adaptively determine the compression ratios for heads and layers across different groups. Compared to the full-size KV cache, our method achieves a compression rate of 4.74%, with a throughput increase of 6.4× and a 1.4× inference speedup in a single batch, incurring only a 1.41% performance loss.

3. Our method maintains performance even under extreme compression rates where some heads are removed. In the needle-in-a-haystack task, with a 9.38% compression rate (viz. compressed size over original size), our method even surpasses the performance of the full-size KV cache version.

## 2 RELATED WORK

### 2.1 ATTENTION-BASED TOKEN EVICTION POLICIES

Early works identify interesting attention patterns in long-context settings, such as the *attention sink* phenomenon (Liu et al., 2024c; Xiao et al., 2023), where models aggregate information using tokens at the beginning and end. Additionally, certain parameters in LLMs remain in an active state (Sun et al., 2024), inspiring researchers to explore the eviction of input prompts and generated tokens. Recent methods employ cumulative attention scores for token eviction strategies (Zhang et al., 2024c; Li et al., 2023; Jiang et al., 2024; Zhang et al., 2024d; Ge et al., 2023; Sheng et al., 2023; Liu et al., 2024d; Li et al., 2024). These methods aim to optimize memory usage while preserving performance. However, they overlook compressing hidden states during the prefilling stage, which is often the most time-consuming aspect. Further exploration is needed to leverage the sparsity of hidden states to optimize compression rates and eviction strategies at this stage.

### 2.2 COMPRESSION OF THE KV CACHE HEADS

Recent studies show that Multi-Head Attention (MHA) varies across heads; some are highly effective at retrieval tasks (Wu et al., 2024), while others specialize in preserving different token types, reflecting inherent patterns of the model (Ge et al., 2023; Jiang et al., 2024). Leveraging the sparsity across these head dimensions allows for designing different KV cache eviction strategies for each head, thereby improving inference speed. Methods like Multi-Query Attention (MQA) (Shazeer, 2019) share keys and values across heads to reduce the KV cache but at the cost of degraded performance. Grouped-Query Attention (GQA) (Ainslie et al., 2023) merges heads and fine-tunes the model, while Multi-Head Latent Attention (MLA) (Liu et al., 2024a) compresses keys and values into a low-rank subspace, reusing the KV cache during inference. However, these methods often require retraining. Yu et al. (2024) use LoRA (Hu et al., 2021) to fine-tune compressed models, minimizing errors in MHA-to-GQA transitions. CHAI (Agarwal et al., 2024) clusters tokens in the prefilling stage but adds significant computational overhead. Efficiently setting compression rates for different heads without training remains an open challenge, requiring further exploration of the sparsity patterns in the heads.

## 2.3 The Compression Behavior of Large Language Models

Sections 2.1 and 2.2 discuss compressing the KV cache by utilizing sparsity patterns of the layer and head dimensions. However, this sparsity pattern is closely related to the model's internal matrix rank and can alter the model's compression behavior (Feng et al., 2022). Recent work (Delétang et al., 2023) reveals that models exhibit spontaneous compression behavior during training, demonstrating that LLMs are powerful general compressors and showcasing the scaling laws of model compression capability. Similar phenomena are observed in (Tao et al., 2024; Huang et al., 2024). Moreover, certain aggregation patterns within prompts in in-context learning are also noted (Wang et al., 2023). These observations inspire us to further explore the internal compression patterns of the model, particularly between different heads and across different layers. We introduce *matrix entropy* (Zhang et al., 2023), which can be considered a measure of the rank of the KV cache; higher entropy indicates more uncertainty and information aggregated by the tokens.

## 3 Method

Inspired by Giraldo et al. (2014), we derive the definition of matrix entropy and introduce our concept of truncated matrix entropy in this section. Additionally, we explore the relationship between matrix entropy and effective rank.

### 3.1 Matrix Entropy

To derive the definition of matrix entropy, we first define the covariance matrix of the model's parameter matrix. In this context, it typically refers to the various heads of the KV cache and the different layers of the hidden state. The covariance matrix, $\Sigma_{\mathbf{X}}$, is derived from the token sequence matrix $\mathbf{X} = [\mathbf{x}_1, \mathbf{x}_2, \ldots, \mathbf{x}_N]$, where $\mathbf{x}_i \in \mathbb{R}^D$ represents the $i$-th token vector in the sequence, and $N$ denotes the sequence length. The covariance matrix $\Sigma_{\mathbf{X}} \in \mathbb{R}^{D \times D}$ is then computed as the average outer product of the centered token vectors:

$$\Sigma_{\mathbf{X}} = \frac{1}{N} \sum_{i=1}^{N} \left( \frac{\mathbf{x}_i - \bar{\mathbf{x}}}{\|\mathbf{x}_i - \bar{\mathbf{x}}\|} \right) \left( \frac{\mathbf{x}_i - \bar{\mathbf{x}}}{\|\mathbf{x}_i - \bar{\mathbf{x}}\|} \right)^T, \tag{1}$$

where $\bar{\mathbf{x}}$ is the mean vector of the sequence:

$$\bar{\mathbf{x}} = \frac{1}{N} \sum_{i=1}^{N} \mathbf{x}_i. \tag{2}$$

It can be shown that $\text{Tr}(\Sigma_{\mathbf{X}}) = 1$, where $\text{Tr}(\cdot)$ represents the trace operator.

Based on the above definition of the covariance matrix, we derive the definition of matrix entropy. Specifically, following Giraldo et al. (2014), the matrix entropy of order $\alpha > 0$ based on $\Sigma_{\mathbf{X}}$ is defined as:

$$S_\alpha(\Sigma_{\mathbf{X}}) = \frac{1}{1 - \alpha} \log \left[ \text{Tr} \left( (\Sigma_{\mathbf{X}})^\alpha \right) \right]. \tag{3}$$

**Lemma 1.** *As $\alpha \to 1$, we obtain the definition of the von Neumann (matrix) entropy (Von Neumann, 2013):*

$$H(\Sigma_{\mathbf{X}}) = -Tr \left( \Sigma_{\mathbf{X}} \log \left( \Sigma_{\mathbf{X}} \right) \right). \tag{4}$$

**Lemma 2.** *Let $\Sigma_{\mathbf{X}}$ be a symmetric positive definite matrix with eigenvalues $\sigma = (\sigma_1, \sigma_2, \ldots, \sigma_D)^T$. The matrix entropy of $\Sigma_{\mathbf{X}}$ can be expressed as:*

$$H(\Sigma_{\mathbf{X}}) = -\sum_{i=1}^{D} \sigma_i \log \sigma_i, \tag{5}$$

where $D$ is the dimension of the covariance matrix of the model's hidden state layer or head. We define matrix entropy on the token sequence and provide the proof in Appendix C. To give an intuitive explanation of its role in the parameter matrix, we introduce effective rank, which links matrix

entropy to dimensionality, such as sequence length and number of heads, leading to meaningful conclusions.

Recent works (Zhang et al., 2023; Zhuo et al., 2023) explore the relationship between matrix entropy and effective rank (Roy & Vetterli, 2007). Zhuo et al. (2023) discuss dimensional collapse and use effective rank to explain asymmetric contrastive learning. Inspired by this, we use effective rank, i.e., matrix entropy, to measure uncertainty across heads and layers, associating higher matrix entropy with more information per token. We remark here that **contrary to the conventional perception that a higher rank represents more information and is less ready for truncation, we observe that to the opposite, as the layer depth increases, since each token becomes more informative by itself, we can thereby directly discard more tokens without hurting the overall information content.** This novel observation is justified by *i)*: Previous efforts (Wang et al., 2023) find that higher-layer tokens gather more information, and a small number of tokens can represent the entire sequence. *ii)*: We observe the same phenomenon in Figure 1: the higher the layer, the higher the matrix entropy of the entire sequence, which means that each token is more informative. *iii)*: **For heads on the same layer, those with a higher effective rank should evict fewer tokens because this head is more informative.** *iv)*: Tokens of the same head in different layers gradually share information as the layers deepen, while tokens of different heads do not share information as the layers deepen. Based on effective rank, we set different compression rates for different heads and layers. The effective rank of $\Sigma_{\mathbf{X}}$, denoted erank($\Sigma_{\mathbf{X}}$), is defined as:

$$\text{erank}(\Sigma_{\mathbf{X}}) = \exp(H(\Sigma_{\mathbf{X}})). \tag{6}$$

**Lemma 3.** *The rank of the covariance matrix $\Sigma_{\mathbf{X}}$ is upper bounded by the rank of the input matrix* $\mathbf{X}$:

$$rank(\Sigma_{\mathbf{X}}) \leq rank(\mathbf{X}). \tag{7}$$

**Lemma 4.** *Eq. 6 can be interpreted as the dimension of the affine subspace spanned, i.e., the effective dimensionality of the parameter matrix in the head and layer dimensions. The bounds are:*

$$1 \leq erank(\Sigma_{\mathbf{X}}) \leq rank(\Sigma_{\mathbf{X}}) \leq D. \tag{8}$$

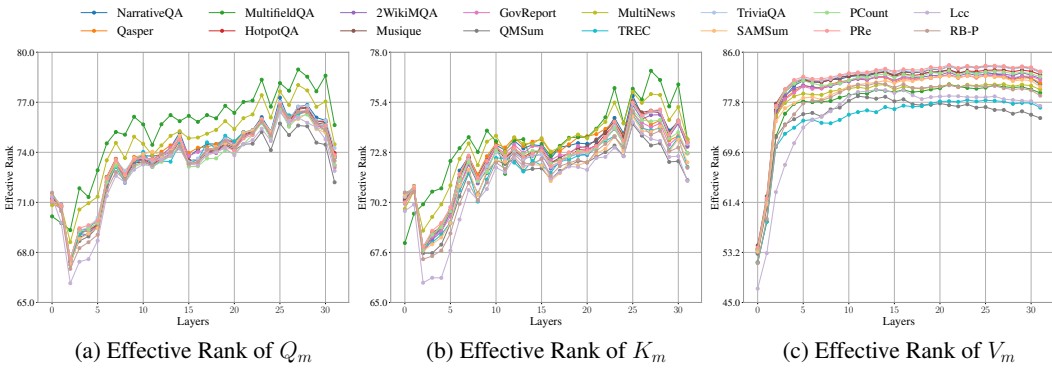

(a) Effective Rank of $Q_m$       (b) Effective Rank of $K_m$       (c) Effective Rank of $V_m$

Figure 1: Effective ranks for the $Q_m$, $K_m$, and $V_m$ across three different datasets in LongBench (Bai et al., 2023) and various layers.

We consider erank($\Sigma_{\mathbf{X}}$) as a measure of the model's uncertainty. In this work, we extend this concept to quantify the model's uncertainty in token sequence representations and derive an uncertainty measure for both the KV cache and the hidden states.

### 3.2 TRUNCATED MATRIX ENTROPY

To identify compression patterns in the hidden states and KV cache, we use matrix entropy. A key question is which matrix—key ($K_m$), query ($Q_m$), or value ($V_m$)—best captures the compression patterns in the token sequences mapped from the hidden states. To investigate this, we plot the matrix entropy trends for $Q_m$, $K_m$, and $V_m$ across different layers and datasets (Figure 1). In our analysis, entropy fluctuations indicate ongoing compression and aggregation within the model layers. Figure 1 reveals: *i)* $Q_m$ and $K_m$ show a stronger compression trend than $V_m$. *ii)* $Q_m$ and $K_m$ have similar effective rank variation, making them equivalent in measuring the KV cache's

compression rate. *iii*) $Q_m$ generally has a larger effective rank compared to $K_m$. Based on Zhuo et al. (2023), a larger rank implies less representation collapse. Thus, we select $Q_m$ to measure effective rank, using randomly sampled data to estimate the matrix entropy of the parameter matrix before testing.

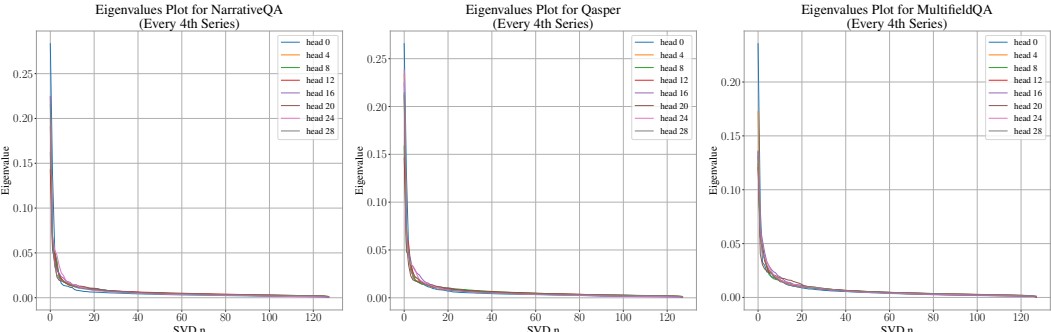

Figure 2: The eigenvalue distribution across three datasets in LongBench and various heads.

We visualize the eigenvalue distribution of $Q_m$ across different heads in the model's final layer. As the distributions show minimal differences across layers, we focus on the last layer over 128 hidden dimensions. Figure 2 reveals: *i*) The initial part of the eigenvalue distribution varies significantly. *ii*) Eigenvalue distributions across heads differ significantly in the leading part of the distribution. *iii*) Different datasets exhibit a similar long-tailed distribution. These findings suggest some heads operate in limited dimensions, suffering from representation collapse, motivating the use of varying compression rates during inference.

**According to Lemma 2, we require $\Sigma_{Q_m}$ to be a positive definite matrix, though it is typically positive semi-definite. To address this, we compute the effective rank using a submatrix because it is positive definite.** Intuitively, to exclude collapsed dimensions when calculating matrix entropy, we adopt a PCA-like method, selecting the top-$k$ eigenvalues before the elbow point (Thorndike, 1953) to determine the effective rank, which represents the entropy of the low-rank matrix. Thus, we introduce the concept of truncated matrix entropy, obtaining the effective rank of a positive definite submatrix to quantify uncertainty. From Eq. 9 and Eq. 6, the truncated effective rank of matrix $Q_m$ is defined by its top $k$ eigenvalues:

$$H_k(\mathbf{\Sigma}_{Q_m}) = -\sum_{i=1}^{k} \sigma_i \log \sigma_i, \tag{9}$$

$$\mathrm{erank}_k(\mathbf{\Sigma}_{Q_m}) = \exp\left(H_k\left(\mathbf{\Sigma}_{Q_m}\right)\right), \tag{10}$$

where $H_k(\mathbf{\Sigma}_{Q_m})$ denotes the entropy calculated using the top-$k$ eigenvalue vector $\sigma$ of the matrix $\mathbf{\Sigma}_{Q_m}$. With the definition of $\mathrm{erank}_k(\mathbf{\Sigma}_{Q_m})$ established, we apply it to the different heads of $Q_m$ across various layers.

### 3.3 UNCERTAINTY-AWARE COMPRESSION STRATEGY

With the aforementioned conclusions, we begin to apply them to compress the heads of the KV cache and each layer of the hidden state. Given that the hidden state and KV cache exhibit similar matrix entropy behavior, we similarly use matrix entropy to adaptively estimate their compression rates. We consider $\mathrm{erank}_k(\mathbf{\Sigma}_{\mathbf{X}})$ as a measure of uncertainty for these matrices. The higher the value, the greater the uncertainty, which implies more information in the token sequence matrix and a lower compression rate for layer and a higher compression rate for head. We present the workflow of our method in Figure 3.

**Inter-layer Compression**  Regarding inter-layer compression, we focus on compressing hidden states $H_m$, used to generate $Q_m$, $K_m$, and $V_m$. This involves measuring model uncertainty of hidden states during the prefilling stage to obtain sparse attention. Unlike previous work (Xu et al., 2023; Li et al., 2023), which prunes the input prompt with a retrieval method before the prefilling stage, we don't require an additional retrieval model, and pruning is conducted internally. Previous work computes all hidden states during the prefilling stage and generates the complete KV cache

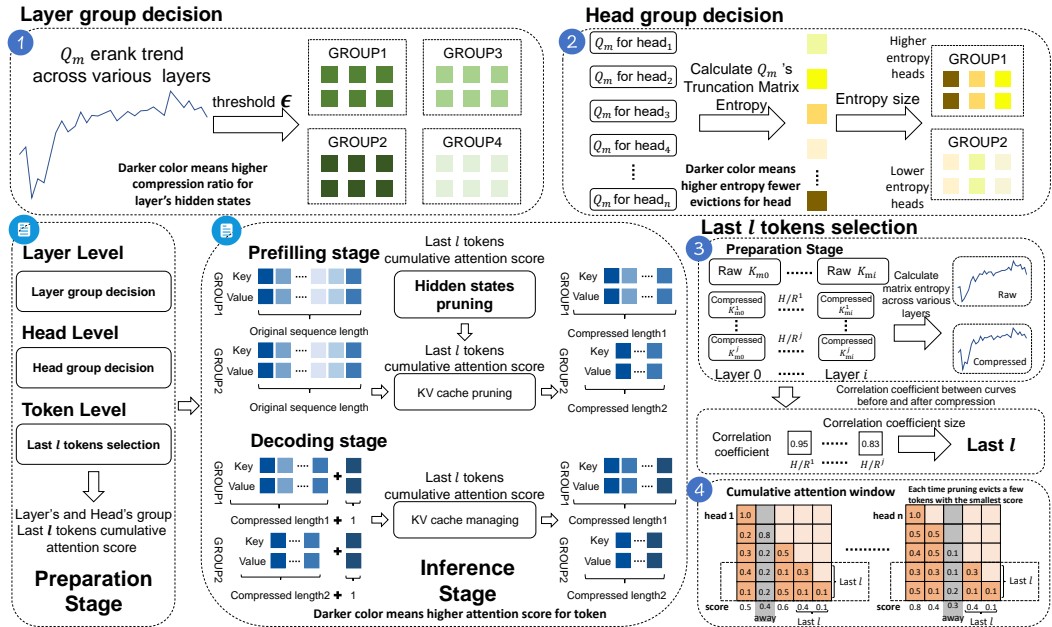

Figure 3: Overview of *UNComp* method. During the preparation stage, before inference on all datasets, we use a small amount of data in Wikitext2 (Merity, 2016) to group the model's layers and heads. Layer grouping is based on Inter-layer Compression, while attention head grouping is determined by Inter-head Compression. At the token level, we determine the cumulative attention score by comparing matrix entropy trends before and after $K_m$ compression (Paragraph 4.2). In the inference stage, compression is guided by the groupings from the preparation phase, and the KV cache size is dynamically managed. The darker the color, the higher the effective rank of the $Q_m$ of the head, and thereby fewer tokens can be evicted. The $Q_{mi}$ refers to the query matrix at layer $i$ before compression. The $H/R^j$ (Paragraph 4.2) refers to the ratio of the number of historical tokens ($H$) to the number of recent tokens ($R$), where $j$ represents the $j$-th ratio in different $H/R$ ratio combination. The $K_{mi}^j$ means the key matrix sampled at layer $i$ with $H/R^j$. Correlation coefficient refers to the Pearson correlation coefficient (Cohen et al., 2009).

before evicting them. Our method differs: we perform eviction on hidden states before generating the KV cache, and the compressed $H_m$ are used to generate the KV cache.

We divide the hidden states of tokens from all layers into $C$ groups, where the token length in each group is consistent across layers. As the group number increases, $\text{erank}_k(\mathbf{X})$ increases, and more tokens are pruned. By observing Figure 1(a), we can identify some patterns. We determine whether to perform compression at two layers where $\text{erank}_k(\mathbf{X})$ decreases beyond a threshold $\epsilon$. This results in the total number of compression stages, $C$:

$$C = \sum_{i=1}^{n-1} \mathbf{1}\left(\text{erank}_k(\mathbf{\Sigma}_{H_m}^{(i)}) - \text{erank}_k(\mathbf{\Sigma}_{H_m}^{(i+1)}) > \epsilon \,\wedge\, \text{erank}_k(\mathbf{\Sigma}_{H_m}^{(i)}) > \text{erank}_k(\mathbf{\Sigma}_{H_m}^{(i+1)})\right), \quad (11)$$

where $\mathbf{1}$ is the indicator function that evaluates to 1 if the decrease in effective rank between layer $i$ and $i + 1$ exceeds the threshold $\epsilon$, and 0 otherwise, and $n$ represents the total number of layers in the model. Eq. 11 is a partition function that determines the division of the model's layers into $C$ groups. The context size at each subsequent group is calculated as follows:

$$S_{i+1} = S_i + \Delta s, \quad i = 1, 2, \ldots, C - 1, \quad (12)$$

$$\Delta s = \frac{S_{\max} - S_{\min}}{C - 1}, \quad (13)$$

where $S_{\max} = S_1$, $S_{\min} = S_C$. $\Delta s$ represents the incremental increase in context size between consecutive groups.

After obtaining the specific compression rate for each layer group, we start to evict tokens from group 2. The hidden states of the tokens in group 1 are usually full-size because in the initial layer,

the matrix entropy is typically small, and we want to retain as much information as possible. From group 2 to group $C$, for all layers, we choose the attention distribution of the final tokens from the previous layer in the prefilling stage and evict the hidden states of the tokens with lower attention scores. In other words, we use the attention scores of the current layer to predict the tokens to be evicted in the next layer.

After generating all the hidden states for the prefilling stage at each layer, we map $H_m$ onto the three matrices $Q_m$, $K_m$, and $V_m$, and during the decoding phase, we maintain a fixed cumulative attention score window, denoted as $w_h$. The window size is $S_{i,h}$, which we will introduce in the next part. Each time a new token is generated, the token with the smallest cumulative attention score in $w_h$ is discarded.

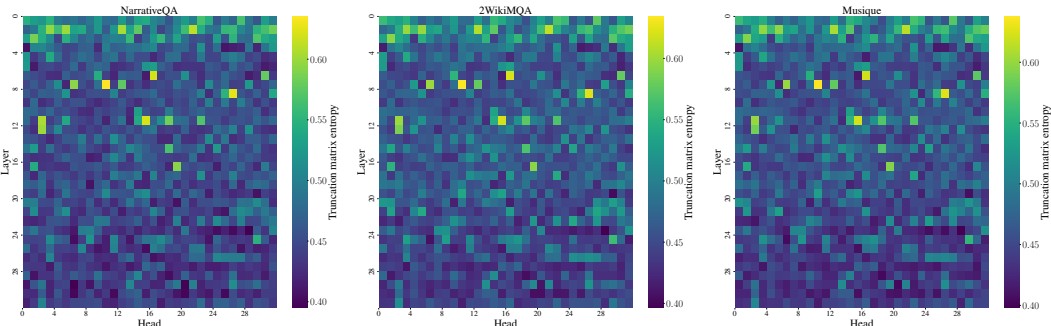

Figure 4: The heatmap of $\mathrm{erank}_k(\Sigma_{Q_m})$ across different layers and heads.

**Inter-head Compression**  Similar to previous works like MQA, GQA, and MLA, we estimate the compression rate in the head dimension. However, unlike their approaches, our method focuses solely on the training-free setting, and we apply different compression rates to different head groups.

We sample 500 data points for observation, typically not included in the current test set, and find that the head patterns across different layers are consistent across datasets, as shown by the effective rank distribution in Figure 4 (More details presented in Appendix A.4). Therefore, we conclude that the effective rank of the heads is not data-dependent, as it is an intrinsic characterization of the model's uncertainty measures. We rank the heads within each layer based on their effective rank and divide them into $m$ groups. The groups are then arranged according to the ranking of the truncated effective rank of each group. Typically, the larger the effective rank, the more information the head contains, the greater the compression ratio, and the fewer tokens are evicted.

After determining the context window $S_i$ from Eq. 12 for each layer, we set different compression rates for head groups based on their truncated effective rank. Specifically, we set the step size $\Delta s_h$, decreasing the context size from $S_{i,1}$, which has the largest effective rank, down to $S_{i,m}$ as follows:

$$S_{i,h} = S_{i,h-1} - (h-1) \cdot \Delta s_h, \quad h = 1, 2, \ldots, m, \tag{14}$$

where $S_{i,h}$ represents the context size for the $h$-th head group at layer $i$, and $\Delta s_h$ is the fixed step size applied between consecutive groups. During decoding, each head maintains its context window according to the compression rate of the group it belongs to. For different heads, we maintain a window of size $S_{i,h}$ using the cumulative attention scores of the last $l$ tokens.

Additionally, we investigate scenarios with extreme compression rates where certain groups with low effective rank are completely removed, excluding them from the forward process. In such cases, the dimensions of the modified attention output are dictated by the altered key and value, leading to a dimensional mismatch with the original attention output. To resolve this, we compute the cosine distance between the attention distributions of the removed and retained heads. These distributions are based on the cumulative attention over the last $l$ tokens. We then fill the attention output matrix with the attention distribution from the head most similar to the removed head's distribution, ensuring alignment with the dimension of the original attention output.

## 4 EXPERIMENT

### 4.1 EXPERIMENTAL SETTINGS

**Models, Baselines, and Tasks**   We evaluate three models: Llama2-7B/13B-chat-hf (Touvron et al., 2023), Llama-3-8B-Inst (AI, 2024), and Mistral-7B-Instruct-v0.1 (Jiang et al., 2023). Llama2 and Llama3 are optimized for dialogue and question-answering, while Mistral is instruction-tuned for similar tasks. We compare our method with existing KV cache eviction methods H2O (Zhang et al., 2024d), PyramidKV (Zhang et al., 2024c), SnapKV (Li et al., 2024), and the head pruning method CHAI (Agarwal et al., 2024) under the same compression rate. Details are in Appendix A. *UNComp* is tested on 16 LongBench (Bai et al., 2023) benchmarks, including 2WikiMQA, GovReport, NarrativeQA, HotpotQA, Musique, Qasper, QMSum, MultiNews, MultifieldQA, TriviaQA, SAMSum, TREC, PCount, PRe, Lcc, and RB-P. We also compare our model's accuracy, inference speed, and throughput under extreme compression and evaluate it on the 'Needle in a Haystack' task (Liu et al., 2024c).

| Methods | Single-Document QA | | | Multi-Document QA | | | Summarization | | | Few-shot Learning | | | Synthetic | | Code | | Avg. | Time (s / sample) |
|---|---|---|---|---|---|---|---|---|---|---|---|---|---|---|---|---|---|---|
| | NtrvQA | Qasper | MF-en | HotpotQA | 2WikiMQA | Musique | GovReport | QMSum | MultiNews | TREC | TriviaQA | SAMSum | PCount | PRe | Lcc | RB-P | | |
| **Llama2-7B-chat-hf, KV Size = FULL** | | | | | | | | | | | | | | | | | | |
| FullKV | 19.34 | 18.61 | 35.19 | 30.66 | 28.42 | 10.05 | 25.19 | 20.18 | 25.73 | 63.00 | 83.62 | 41.60 | 5.00 | 10.00 | 61.40 | 55.45 | 33.34 | 0.86 |
| **Llama-2-7B-chat-hf, KV Size = 384, Compressibility is 9.38% (Except CHAI method)** | | | | | | | | | | | | | | | | | | |
| H2O | 14.96 | 14.60 | 17.40 | 26.72 | 27.97 | 6.11 | 17.83 | 18.76 | 20.17 | 47.00 | 77.56 | **39.39** | 4.50 | 5.00 | 57.08 | 50.31 | 27.84 | 0.94 |
| SnapKV | 16.27 | 17.34 | 30.37 | 33.04 | 27.82 | 9.92 | 19.34 | 20.33 | 22.63 | 59.50 | 83.50 | 38.45 | **5.50** | **12.50** | 59.18 | **55.28** | 31.94 | 0.82 |
| Pyramidkv | 16.86 | 18.26 | 31.01 | **31.59** | 27.93 | 8.69 | 19.88 | 20.15 | 22.43 | 62.00 | 83.86 | 38.98 | **5.50** | 10.00 | 58.94 | 52.80 | 31.81 | 0.84 |
| CHAI | 16.75 | 16.91 | **34.69** | 26.09 | 20.80 | 9.20 | **20.79** | 20.23 | 23.33 | 57.00 | 75.52 | 35.67 | 4.00 | 6.33 | 50.10 | 46.55 | 29.00 | 1.51 |
| Ours-group-stage | **17.61** | **20.39** | 33.56 | 30.52 | 26.75 | 9.91 | 20.42 | **20.55** | 23.54 | **63.00** | 82.51 | 38.16 | 4.50 | 8.00 | 59.76 | 52.55 | 31.98 | **0.61** |
| Ours-group | 17.33 | 19.34 | 34.16 | 31.54 | **28.23** | **10.04** | 20.38 | 20.51 | 23.33 | **63.00** | **84.11** | 39.35 | 5.50 | 9.50 | **59.93** | 54.87 | **32.57** | 0.81 |
| **Llama2-13B-chat-hf, KV Size = FULL** | | | | | | | | | | | | | | | | | | |
| FullKV | 18.20 | 26.07 | 37.06 | 36.20 | 32.44 | 14.19 | 25.82 | 20.20 | 26.00 | 66.50 | 87.49 | 35.93 | 3.12 | 11.50 | 53.29 | 52.73 | 34.17 | 2.01 |
| **Llama-2-13B-chat-hf, KV Size = 384 , Compressibility is 9.38% (Except CHAI method)** | | | | | | | | | | | | | | | | | | |
| H2O | 14.11 | 18.36 | 22.78 | 33.03 | 27.58 | 12.94 | 18.97 | 18.69 | 20.37 | 53.50 | 85.75 | 34.15 | 3.55 | 6.00 | 50.97 | 47.56 | 29.27 | 2.57 |
| SnapKV | 17.09 | 22.77 | 34.37 | 36.73 | **31.04** | 13.02 | 19.70 | 20.00 | 22.91 | 63.00 | 87.48 | **37.44** | 4.05 | 11.50 | 51.76 | 51.27 | 32.70 | 1.93 |
| PyramidKV | 16.33 | 22.81 | 34.19 | **37.54** | 30.25 | 13.82 | 19.79 | **20.11** | 23.14 | 64.50 | 86.45 | 36.62 | 4.05 | **12.00** | **52.06** | 50.58 | 32.77 | 3.50 |
| CHAI | 17.06 | 23.51 | 31.01 | 33.70 | 27.78 | 11.73 | **23.03** | 19.53 | **24.66** | 63.00 | 86.18 | 15.93 | 4.00 | 8.50 | 45.57 | 48.74 | 30.37 | 2.50 |
| Ours-group-stage | 15.20 | 23.03 | 35.44 | 36.66 | 30.21 | 12.67 | 20.70 | 19.53 | 24.05 | 63.50 | 85.10 | 35.71 | 3.65 | 9.50 | 49.76 | 46.78 | 31.97 | **1.74** |
| Ours-group | **18.16** | **23.90** | **36.56** | 36.29 | 30.48 | **14.36** | 21.22 | 19.93 | 24.06 | **67.00** | **88.11** | 36.02 | 4.00 | 11.50 | 51.59 | **51.53** | **33.42** | 1.85 |
| **Llama3-8B-Instruct, KV Size = FULL** | | | | | | | | | | | | | | | | | | |
| FullKV | 23.31 | 31.18 | 38.09 | 43.67 | 35.26 | 21.43 | 28.42 | 22.9 | 26.64 | 73.5 | 89.76 | 42.2 | 4.78 | 67.88 | 60.12 | 56.76 | 41.62 | 2.88 |
| **Llama-3-8B-Instruct, KV Size = 384 , Compressibility is 4.74% (Except CHAI method)** | | | | | | | | | | | | | | | | | | |
| H2O | 18.80 | 13.76 | 21.20 | 38.90 | 31.38 | 14.81 | 20.38 | 20.70 | 22.03 | 61.00 | 82.07 | 39.49 | **5.12** | 66.92 | 58.59 | 54.98 | 35.63 | 3.98 |
| SnapKV | 21.47 | 19.77 | 33.97 | 43.10 | 32.79 | **21.48** | 21.69 | 22.01 | 22.92 | 63.00 | 89.69 | 39.78 | 5.06 | 67.83 | 60.19 | 56.82 | 38.85 | 2.68 |
| Pyramidkv | 22.08 | 19.43 | 32.99 | 42.51 | 32.01 | 19.62 | 21.73 | 22.24 | 22.74 | **71.00** | 89.59 | **40.51** | 4.23 | **68.50** | 58.92 | 53.92 | 38.88 | 2.78 |
| CHAI | 18.99 | 23.44 | 31.82 | 33.37 | 22.63 | 19.07 | **24.46** | 21.74 | 23.78 | 71.00 | 89.28 | 37.15 | 4.92 | 67.75 | 44.44 | 36.12 | 35.50 | 4.20 |
| Ours-group-stage | 21.08 | **24.60** | 33.87 | 44.07 | 33.72 | 20.42 | 22.12 | 21.76 | **24.06** | 71.00 | **91.11** | 40.07 | 4.43 | 60.50 | **62.10** | 57.36 | 39.52 | 2.65 |
| Ours-group | **22.85** | 24.27 | **35.32** | **44.30** | **34.42** | 20.46 | 22.25 | **22.25** | 23.74 | 71.00 | 89.64 | 40.43 | 4.56 | 68.00 | 61.71 | **58.13** | **40.21** | **2.45** |
| **Mistral-7B-Instruct, KV Size = FULL** | | | | | | | | | | | | | | | | | | |
| FullKV | 20.53 | 27.28 | 47.67 | 38.57 | 26.75 | 15.46 | 30.88 | 22.00 | 26.99 | 70.50 | 86.28 | 43.23 | 1.30 | 29.42 | 56.60 | 52.18 | 37.22 | 2.74 |
| **Mistral-7B-Instruct, KV Size = 384 , Compressibility is 9.38% (Except CHAI method)** | | | | | | | | | | | | | | | | | | |
| H2O | 14.68 | 14.92 | 28.43 | 30.04 | 20.92 | 11.21 | 20.03 | 19.37 | 21.05 | 58.00 | 81.48 | 41.08 | **2.42** | 11.04 | 54.76 | 48.78 | 29.89 | 3.13 |
| SnapKV | 19.09 | 23.06 | 46.73 | 35.76 | **25.41** | **15.37** | 23.23 | 21.61 | 23.78 | 64.00 | 85.34 | 41.79 | 1.09 | **28.92** | 55.46 | **51.52** | 35.14 | 2.55 |
| Pyramidkv | 17.59 | 22.82 | 46.34 | 36.01 | 25.07 | 14.47 | 21.83 | 21.83 | 22.83 | 69.00 | 85.85 | 42.42 | 1.67 | 27.98 | 53.32 | 49.03 | 34.93 | 2.57 |
| CHAI | 15.01 | 17.23 | 39.41 | 23.53 | 21.12 | 9.00 | 24.83 | 20.59 | 23.17 | 49.50 | 84.08 | 33.28 | 1.07 | 19.25 | 46.85 | 44.62 | 29.53 | 3.55 |
| Ours-group-stage | **19.65** | 21.93 | 44.08 | 35.30 | 23.21 | 12.84 | 24.61 | **22.08** | **24.36** | 67.00 | 84.31 | 41.81 | 1.48 | 28.81 | 54.84 | 52.10 | 34.90 | **2.27** |
| Ours-group | 18.98 | **23.30** | **48.77** | **36.80** | 24.83 | 15.22 | **24.87** | 21.65 | 23.78 | **69.50** | **85.93** | **43.17** | 0.94 | 27.38 | 55.28 | 50.91 | **35.71** | 2.79 |

Table 1: Performance Comparison across Different Tasks: Ours-group-stage compresses both hidden states and KV cache, while Ours-group compresses only the KV cache. For 150 data points, Ours-group-stage is 1.4x faster than Ours-group, with only a 0.59% performance loss. All methods, except CHAI, compress key and value caches at the same compression ratio. In contrast, CHAI primarily compresses the key cache, leaving the value cache uncompressed and performing selection at the attention head level, achieving a 77.54% compression ratio. The last column represents the average time per sample.

### 4.2 EXPERIMENTAL COMPARISON IN MEMORY-CONSTRAINED SETTING

**Main Results**   Our main experimental results are shown in Table 1, where we present the performance of current mainstream LLMs tested on LongBench. We perform the comparison under a unified setting with a KV cache size of 384. The heads in the main table are divided into two groups, with KV cache sizes set to 512 and 256, respectively, in descending order of effective rank. To ensure a fair comparison with others, we set the KV cache size of the other models to 384.

From Table 1, we draw the following conclusions: *i)* Our method achieves the best performance on LLaMA3, delivering a speedup of up to 1.4 times on a single batch instance with a compression

| Llama2-7B-chat-hf, KV Size = 64 | | | | | | | |
|---|---|---|---|---|---|---|---|
| **Methods** | **Qasper** | **Musique** | **GovReport** | **TREC** | **PCount** | **Lcc** | **Average score** |
| FullKV | 18.61 | 10.05 | 25.19 | 63.00 | 5.00 | 61.40 | 30.54 |
| H2O | 13.84 | 1.33 | 8.57 | 18.00 | 0.50 | 28.86 | 11.85 |
| PyramidKV | 16.10 | 6.58 | 12.07 | 46.00 | **5.50** | 46.09 | 22.06 |
| SnapKV | 15.70 | 6.15 | 11.16 | 40.50 | 5.00 | 43.77 | 20.38 |
| Ours-group-stage | **17.57** | 6.58 | **15.25** | **59.00** | 4.50 | 49.75 | **25.44** |
| Ours-group | 15.40 | **6.96** | 14.88 | 55.50 | 5.00 | **50.00** | 24.62 |
| **Llama2-7B-chat-hf, KV Size of One Group With Extreme Compression** | | | | | | | |
| **Methods** | **Qasper** | **Musique** | **GovReport** | **TREC** | **PCount** | **Lcc** | **Average score** |
| FullKV | 18.61 | 10.05 | 25.19 | 63.00 | 5.00 | 61.40 | 30.54 |
| Ours-remain-tokens-256 | 19.67 | 9.80 | 20.19 | 63.00 | 5.50 | 60.28 | 29.74 |
| Ours-remain-tokens-128 | 18.67 | 9.75 | 20.02 | 63.00 | 5.50 | 59.60 | 29.42 |
| Ours-remain-tokens-64 | 18.13 | 9.79 | 19.84 | 63.00 | 5.50 | 58.09 | 29.06 |
| Ours-remain-tokens-32 | 18.04 | 9.24 | 19.31 | 63.00 | 5.50 | 57.04 | 28.69 |
| Ours-remain-tokens-16 | 18.48 | 8.08 | 18.21 | 63.00 | 5.00 | 47.29 | 26.68 |
| Ours-remain-tokens-12 | 17.31 | 8.78 | 18.16 | 62.00 | 5.00 | 45.23 | 26.08 |
| Ours-delete-2-heads | 18.30 | 8.58 | 18.98 | 63.00 | 5.50 | 59.38 | 28.96 |
| Ours-delete-4-heads | 13.29 | 7.96 | 19.12 | 62.50 | 5.50 | 53.94 | 27.05 |
| Ours-delete-8-heads | 12.64 | 6.60 | 9.87 | 63.50 | 3.21 | 37.87 | 22.28 |

Table 2: Extreme Compression Conditions.

rate of 4.68%, confirmed by averaging multiple repeated experiments. *ii)* Our method exhibits near-lossless performance in certain models, especially compared to the full-size KV cache setting in Llama2-7B/13B-chat-hf, with only a 0.77% performance loss while achieving a 9.38% compression rate. *iii)* In comparison with the training-free head-pruning method CHAI, our *UNComp* outperforms in both single-batch inference speed and overall performance. With a KV cache compression rate lower than CHAI's 68.55%, our method achieved 5.4 times faster inference speed.

**The Result of Extreme Compression**   We compare performance under extreme compression settings to highlight our method's advantages. For this investigation, we use Llama-2-7B-chat-hf as the baseline model, categorizing the layer into five groups and attention heads into two groups. As shown in Table 2, we evaluate the effectiveness of our approach across increasingly extreme compression ratios. Importantly, when the compression rate of the KV cache is set to 1.56%, our method shows a substantial enhancement over existing alternatives.

We further explore the minimum achievable compression rate for the group with the lower effective rank, as detailed in Table 2 where *Ours-remain-tokens-N* indicates the retention of $N$ tokens per attention head within the group of lower effective rank, while the other group maintains a KV cache size of 512. Furthermore, *Ours-delete-K-head* denotes the complete pruning of $K$ heads per layer, contingent on the effective rank order. **The results underscore that our methodology can sustain a high level of accuracy relative to the full KV cache size when only 12 tokens are preserved or even certain heads are pruned.** This finding further corroborates the validity of employing differentiated compression ratios for various heads, aligned with their respective effective ranks.

| **Methods** | **NVIDIA A100 80GB GPU** | | | | **AMD Instinct MI210 64G GPU** | | | |
|---|---|---|---|---|---|---|---|---|
| | **Attention Time (s)** | **Prefill Time (s)** | **Decoding Time (s)** | **Max Memory Usage (MB)** | **Attention Time (s)** | **Prefill Time (s)** | **Decoding Time (s)** | **Max Memory Usage (MB)** |
| FullKV | 129.13 | 77.34 | 51.79 | 25900 | 189.92 | 116.64 | 73.27 | 23195 |
| H2O | 140.92 | 90.56 | 50.37 | **22908** | 241.42 | 173.08 | 68.34 | **20247** |
| PyramidKV | 126.03 | 78.98 | 47.05 | 22936 | 185.85 | 119.17 | 66.68 | 20295 |
| SnapKV | 123.71 | 78.71 | 45.00 | 22920 | 184.59 | 120.26 | 64.33 | 20276 |
| Ours-group-stage | **91.56** | **48.78** | 42.78 | 22964 | **155.60** | **100.17** | **55.43** | 20300 |
| Ours-group | 121.85 | 79.60 | **42.25** | 22978 | 184.76 | 121.04 | 63.72 | 20335 |

Table 3: Single Batch Time Consumption and Memory Usage Analysis on Different GPUs. Analysis based on MultifieldQA dataset with 150 samples, and KV size is 384 per layer. The compression ratio of Ours-group-stage to hidden states in the prefill stage is 63.09%.

**Analysis of Inference Time Latency and Performance**   We analyze the inference time latency and the specific time costs of each component. To achieve reliable time analysis, we synchronize the CPU and GPU clock frequencies to facilitate our measurements. We use the Llama2-7B-chat-hf model to measure 150 data points from the MultifieldQA collection in a single batch on an NVIDIA A100 80G GPU and an AMD INSTINCT MI210 64G GPU. We focus on the duration of the prefill-

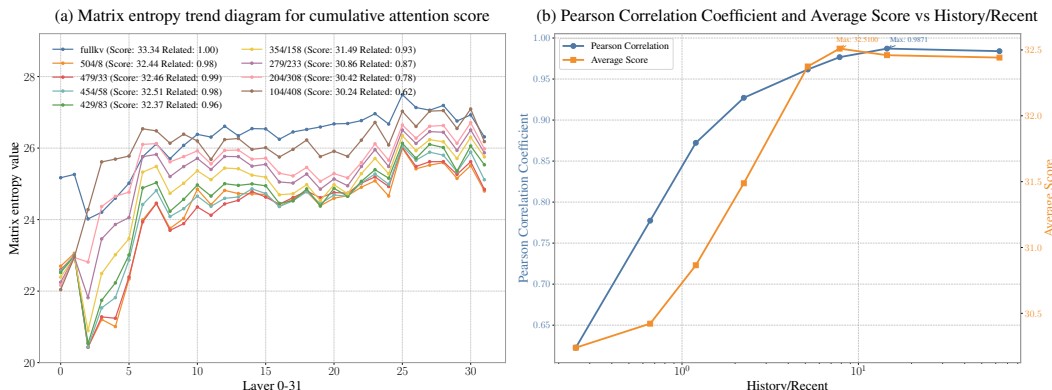

Figure 5: Comparison of Inter-Layer Matrix Entropy Trends Across Different $H/R$.

ing stage, the decoding duration, and the total duration of the attention mechanism. We also record the maximum memory usage during runtime.

In Table 3, we compare our experimental results and present the following observations: *i)* For long-context generation tasks, the prefilling stage takes up more time throughout the inference process. *ii)* Our method is optimized in the prefilling stage, greatly speeding up inference and outperforming other methods, with up to $1.58\times$ acceleration over the full-size KV cache in a single batch. For throughput analysis, experiments with prompt and generation lengths set to $2048 + 8096$ show that FullKV supports a maximum batch size of 6 with a token generation time of 15.67 ms. In contrast, our method supports a batch size of 32 with a token generation time of 2.45 ms, achieving 6.4 times the throughput of FullKV. More details are provided in Appendix E.

**Needle in a Haystack Task** The 'Needle in a Haystack' task (Liu et al., 2024c) involves embedding key information randomly in long contexts to assess the model's ability to handle complex, extended text. Table 4 compares Llama2-4k and Llama3-8k, both with a KV size of 384. The results show that our method outperforms FullKV at a 9.38% compression rate, demonstrating its superiority. This indicates that our uncertainty measurement method based on effective rank can identify the heads crucial for the retrieval task and effectively compress noisy heads.

| Methods | Llama2-4k | Llama3-8k |
|---|---|---|
| FullKV | 98.70 | 84.99 |
| H2O | 61.14 | 51.56 |
| PyramidKV | 93.24 | 79.08 |
| SnapKV | 94.50 | 81.27 |
| CHAI | 97.80 | 64.69 |
| Ours-group | 98.42 | **84.13** |
| Ours-group-stage | **98.80** | 83.73 |

Table 4: Needle-in-a-haystack results.

**The Ratio of Recent Tokens to Historical Window Tokens** This part examines the ratio $H/R$, comparing the number of historical tokens ($H$) to the most recent $l$ tokens ($R$). Our experiment reveals that matrix entropy trends across layers at various $H/R$ ratios align with those of the full-size KV cache. As shown in Figure 5(a), different proportions of historical and recent tokens produce distinct trends. Notably, a compressed key matrix trend more similar to the full KV cache indicates better performance, given the same KV cache compression ratio. We confirm this through multiple experiments. To measure this similarity, we use the Pearson correlation coefficient. As depicted in Figure 5(b), higher similarity in matrix entropy trends corresponds to improved model performance.

## 5 CONCLUSION

We introduced *UNComp*, an uncertainty-aware approach for compressing the KV cache and hidden states in LLMs. By employing matrix entropy to measure model uncertainty across layers and heads, *UNComp* adaptively determines compression rates, achieving a balance between memory efficiency and model performance. Our experiments demonstrate that *UNComp* can reduce memory usage significantly, achieving up to a $1.6\times$ speedup during the prefilling stage, a $6.4\times$ throughput improvement, and compressing the KV cache to 4.74% of its original size. Despite this high compression, *UNComp* maintains a minimal performance loss of only 1.41%, and even surpasses the performance of the full-size KV cache in specific needle-in-a-haystack tasks. This indicates that our method provides an effective solution for optimizing long-context LLM inference without requiring additional training. Moving forward, our approach can serve as a foundation for further exploration into adaptive compression techniques in large-scale model deployment.

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

# A  IMPLEMENTATION DETAILS

## A.1  MACHINE ENVIRONMENT

Main of our experiments are conducted on eight AMD INSTINCT MI210 64G GPUs. For the time test analysis, we conducted the experiment on a single NVIDIA A100 80GB GPU and a single AMD INSTINCT MI210 64G GPU.

## A.2  MODEL SELECTION

In all of our experiments, the model's weights are downloaded from huggingface. For all llama architectures, Llama2-7b model uses the 'meta-llama/Llama-2-7b-chat-hf' version, Llama2-13b model uses the 'meta-llama/Llama-2-13b-chat-hf' version, Llama3-8b model uses the 'meta-llama/Meta-Llama-3-8B-Instruct' version. For mistral architecture, 'mistralai/Mistral-7B-Instruct-v0.2' version is used.

## A.3  HYPERPARAMETER SETTING

We conduct the experiment in a scenario with an average KV size of 384 per layer. The experiment is governed by six main hyperparameters, the selection of last $l$ token's cumulative attention score, the threshold $\epsilon$, the setting of minimum context length $S_{min}$, the numbers of groups (called $GN$) and the selections of $S_{i,1}$ and $\Delta s_h$.

In our experiment, we select the cumulative attention scores of the last 8 tokens, and threshold $\epsilon$ is set to 1. $S_{min}$ is set to 1536 and $S_{max}$ is determined by the maximum context length of the model. We conduct the experiment under two groups, setting $GN$ to 2. The groups with the higher truncated matrix entropy are assigned a higher compression rate, while the groups with the lower truncated matrix entropy are assigned a lower compression rate. The $S_{i,1}$ is set to 512, and the corresponding $\Delta s_h$ is set to 256.

More specifically, we randomly sample several Wikitext2 datasets and calculate the matrix entropy inter-layer trend of the query matrix in the prefill stage. Based on the method outlined in the paper, we partition the 32 attention layers of Llama2-7B, Mistral-7B, and Llama3-8B into five distinct groups: layers 0–1, 2–14, 15–25, 26–30, and layer 31. The reserved capacities for these groups are configured as 4096, 3456, 2816, 2176, and 1536, respectively, in Llama2-7B and Mistral-7B. For Llama3-8B, the corresponding reserved sizes are set to 8096, 6456, 4816, 3176, and 1536. For Llama2-13B, which consists of 40 attention layers, we categorize the layers into eight groups: layer 0, layers 1–2, 3–14, 15–22, 23–30, 31–34, 35–36, and 37–39, with the reserved sizes designated as 4096, 3731, 3366, 3001, 2636, 2271, 1906, and 1536, respectively.

## A.4  ATTENTION HEAD TYPE SCREENING

In this study, we use the Wikitext2 dataset to categorize attention heads. Taking Llama2-7B-chat-hf as an example, we divide the attention heads into two groups for demonstration purposes.

Initially, we randomly select 500 data samples and input them into a large model for generation. During the prefilling stage, we compute the truncated matrix entropy of the query matrix for each layer's 32 attention heads. Attention heads with higher truncated matrix entropy are assigned a value of 1, while those with lower entropy are assigned a value of 0, and the results are output to a file. Once the data generation process is complete, we collect 500 files containing 32x32 matrices.

In the actual generative task, we load the previously generated file. For each layer, we count the number of 1s per attention head and group the 16 heads with the highest counts together, assigning them a higher compression rate. The remaining 16 attention heads are grouped separately and assigned a lower compression rate.

# B  DETAILS OF EVALUATION

Longbench is the first benchmark for assessing the long-context understanding capabilities of large language models in a bilingual and multitask framework. It evaluates multilingual capabilities in

both Chinese and English, consisting of six major categories and twenty-one tasks. Key application scenarios include single-document QA, multi-document QA, summarization, few-shot learning, synthetic tasks, and code completion. We use Longbench to evaluate the performance of our method on contextual input tasks. The details of metrics at Table 5.

Additionally, once the data sample is encoded into tokens, if its length exceeds the model's maximum allowable length, we truncate it by taking equal portions from the beginning and the end.

| Dataset | Metric | Language | Data Length | Dataset | Metric | Language | Data Length |
|---------|--------|----------|-------------|---------|--------|----------|-------------|
| NarrativeQA | F1 | English | 200 | MultiNews | range-l | English | 200 |
| Qasper | F1 | English | 200 | trec | classification accuracy | English | 200 |
| MultifieldQA | F1 | English | 150 | TriviaQA | English | F1 | 200 |
| HotpotQA | F1 | English | 200 | SAMSum | range-l | English | 200 |
| 2WikiMQA | F1 | English | 200 | PCount | exact match accuracy | English | 200 |
| Musique | F1 | English | 200 | PRe | exact match accuracy | English | 200 |
| GovReport | range-l | English | 200 | Lcc | edit similarity | Python/C#/Java | 500 |
| QMSum | range-l | English | 200 | RB-P | edit similarity | Python/Java | 500 |

Table 5: The details of statistics in LongBench

## C  APPENDIX FOR PROOFS

**Proof Lemma 1**

**Proof.**  To derive the von Neumann entropy from the Rényi entropy, we first need to clarify the relationship between the two. The von Neumann entropy can be seen as a special case of the Rényi entropy in the limit where the Rényi parameter $\alpha \to 1$. The Rényi entropy is defined as:

$$S_\alpha(\Sigma_\mathbf{X}) = \frac{1}{1-\alpha} \log\left(\text{Tr}((\Sigma_\mathbf{X})^\alpha)\right), \tag{15}$$

where $\alpha$ is the order of the Rényi entropy, $\Sigma_\mathbf{X}$ is the density matrix, and $\text{Tr}(\rho^\alpha)$ is the trace of the density matrix raised to the power of $\alpha$. To derive the von Neumann entropy, we need to examine the limit of the Rényi entropy as $\alpha \to 1$. Let's consider the form of the Rényi entropy:

$$S_\alpha(\Sigma_\mathbf{X}) = \frac{1}{1-\alpha} \log\left(\sum_i \sigma_i^\alpha\right), \tag{16}$$

where $\sigma_i$ are the eigenvalues of the density matrix $\Sigma_\mathbf{X}$. As $\alpha \to 1$, we can apply L'Hôpital's rule to compute this limit:

$$S(\Sigma_\mathbf{X}) = \lim_{\alpha \to 1} S_\alpha(\rho) = \lim_{\alpha \to 1} \frac{1}{1-\alpha} \log\left(\sum_i \sigma_i^\alpha\right) \tag{17}$$

To proceed, consider the Taylor expansion of $\sum_i \sigma_i^\alpha$:

$$\sum_i \sigma_i^\alpha = \sum_i \sigma_i \cdot e^{(\alpha-1)\log\sigma_i} \approx \sum_i \sigma_i\left(1 + (\alpha-1)\log\sigma_i\right) = 1 + (\alpha-1)\sum_i \sigma_i\log\sigma_i \tag{18}$$

Thus,

$$S_\alpha(\Sigma_\mathbf{X}) \approx \frac{1}{1-\alpha} \log\left(1 + (\alpha-1)\sum_i \sigma_i\log\sigma_i\right) \tag{19}$$

As $\alpha \to 1$, we can use the approximation $\log(1+x) \approx x$ for small $x$. Therefore, we get:

$$S_\alpha(\mathbf{\Sigma_X}) \approx -\sum_i \sigma_i \log \sigma_i \tag{20}$$

which is exactly the expression for the von Neumann entropy:

$$H(\mathbf{\Sigma_X}) = -\mathrm{Tr}(\mathbf{\Sigma_X} \log(\mathbf{\Sigma_X})) \tag{21}$$

**Proof Lemma 2**

**Proof.** In this section, we present a continuous proof of the transformation from the matrix entropy formula to the eigenvalue form.

$$H(\mathbf{\Sigma_X}) = -\mathrm{Tr}\left(\mathbf{\Sigma_X} \log\left(\mathbf{\Sigma_X}\right)\right) \tag{22}$$

Given that $\mathbf{\Sigma_X}$ is a symmetric positive definite matrix, we can perform an eigenvalue decomposition:

$$\mathbf{\Sigma_X} = \mathbf{U\Lambda U}^\top \tag{23}$$

where $\mathbf{U}$ is an orthogonal matrix composed of eigenvectors, and $\mathbf{\Lambda}$ is a diagonal matrix whose entries are the eigenvalues $\sigma_1, \sigma_2, \ldots, \sigma_D$. The logarithm of $\mathbf{\Sigma_X}$ can then be written as:

$$\log(\mathbf{\Sigma_X}) = \mathbf{U} \log(\mathbf{\Lambda}) \mathbf{U}^\top \tag{24}$$

where $\log(\mathbf{\Lambda})$ is a diagonal matrix whose elements are $\log(\sigma_1), \log(\sigma_2), \ldots, \log(\sigma_D)$. Substituting these into the entropy expression:

$$H(\mathbf{\Sigma_X}) = -\mathrm{Tr}\left(\mathbf{U\Lambda U}^\top \mathbf{U} \log(\mathbf{\Lambda}) \mathbf{U}^\top\right) \tag{25}$$

Since $\mathbf{U}^\top \mathbf{U} = \mathbf{I}$, this simplifies to:

$$H(\mathbf{\Sigma_X}) = -\mathrm{Tr}\left(\mathbf{\Lambda} \log(\mathbf{\Lambda})\right) \tag{26}$$

For a diagonal matrix, the trace is the sum of its diagonal elements. Therefore, we have:

$$H(\mathbf{\Sigma_X}) = -\sum_{i=1}^{D} \sigma_i \log(\sigma_i) \tag{27}$$

This concludes the proof that the matrix entropy formula can be written as the sum of the eigenvalues of $\mathbf{\Sigma_X}$.

**Proof Lemma 3**

**Proof.** Let $\mathbf{X} \in \mathbb{R}^{n \times p}$ be a matrix representing $n$ observations and $p$ variables. The covariance matrix $\mathbf{\Sigma_X}$ of $\mathbf{X}$ is defined as:

$$\mathbf{\Sigma_X} = \frac{1}{n-1} \mathbf{X}^\top \mathbf{X} \tag{28}$$

The goal is to determine the relationship between the rank of the matrix $\mathbf{X}$ and the rank of its covariance matrix $\mathbf{\Sigma_X}$.

The rank of the matrix $\mathbf{X}$, denoted as $\mathrm{rank}(\mathbf{X})$, is the number of linearly independent columns in $\mathbf{X}$, and it satisfies the inequality:

$$\mathrm{rank}(\mathbf{X}) \leq \min(n, p) \tag{29}$$

Since the covariance matrix $\Sigma_{\mathbf{X}}$ is given by $\Sigma_{\mathbf{X}} = \frac{1}{n-1}\mathbf{X}^\top\mathbf{X}$, it is a $p \times p$ symmetric matrix. The rank of $\Sigma_{\mathbf{X}}$, denoted $\text{rank}(\Sigma_{\mathbf{X}})$, is determined by the product $\mathbf{X}^\top\mathbf{X}$. The rank of this product is bounded by the rank of $\mathbf{X}$, so we have the following inequality:

$$\text{rank}(\Sigma_{\mathbf{X}}) \leq \text{rank}(\mathbf{X}) \tag{30}$$

This shows that the rank of the covariance matrix $\Sigma_{\mathbf{X}}$ cannot exceed the rank of the original matrix $\mathbf{X}$. In the case where the number of observations $n$ is greater than or equal to the number of variables $p$ (i.e., $n \geq p$), and the columns of $\mathbf{X}$ are linearly independent, the rank of $\mathbf{X}$ is equal to $p$, meaning $\text{rank}(\mathbf{X}) = p$. In this scenario, the matrix $\mathbf{X}^\top\mathbf{X}$ has full rank, which implies that the covariance matrix $\Sigma_{\mathbf{X}}$ will also have full rank. Therefore, we have $\text{rank}(\Sigma_{\mathbf{X}}) = p$, and the rank of the covariance matrix is equal to the rank of the original matrix, i.e., $\text{rank}(\Sigma_{\mathbf{X}}) = \text{rank}(\mathbf{X})$.

On the other hand, when the number of observations is less than the number of variables (i.e., $n < p$), the rank of $\mathbf{X}$ is constrained by the number of observations, such that $\text{rank}(\mathbf{X}) \leq n$. Consequently, the rank of the covariance matrix $\Sigma_{\mathbf{X}}$ is also limited by $n$, meaning $\text{rank}(\Sigma_{\mathbf{X}}) \leq n$. Since $n < p$ in this case, the covariance matrix is rank-deficient, and we have $\text{rank}(\Sigma_{\mathbf{X}}) < p$.

In general, the rank of the covariance matrix $\Sigma_{\mathbf{X}}$ is less than or equal to the rank of the original matrix $\mathbf{X}$. Specifically, $\text{rank}(\Sigma_{\mathbf{X}}) = \text{rank}(\mathbf{X})$ when the number of observations $n \geq p$ and the columns of $\mathbf{X}$ are linearly independent. However, when $n < p$, the covariance matrix $\Sigma_{\mathbf{X}}$ will be rank-deficient, such that $\text{rank}(\Sigma_{\mathbf{X}}) < p$.

**Proof Lemma 4**

**Proof.** The entropy $H(\Sigma_{\mathbf{X}})$ of a set of singular values $\sigma_1, \sigma_2, \ldots, \sigma_D$ is given by the formula:

$$H(\sigma_1, \sigma_2, \ldots, \sigma_D) = -\sum_{i=1}^{D} \sigma_i \log \sigma_i. \tag{31}$$

The trace of $\Sigma_{\mathbf{X}}$, $\text{Tr}(\Sigma_{\mathbf{X}})$, is 1. Since entropy measures the uncertainty or disorder in a distribution, we can establish certain bounds for the entropy based on the structure of the singular values.

First, we note that if the distribution is concentrated entirely at a single value (i.e., all but one of the singular values are zero), then the entropy will be minimized at 0. Specifically:

$$H(1, 0, \ldots, 0) = 0. \tag{32}$$

On the other hand, the entropy is maximized when the singular values are uniformly distributed. In the case of a uniform distribution over $D$ singular values, we have:

$$\sigma_1 = \sigma_2 = \cdots = \sigma_D = \frac{1}{D}, \tag{33}$$

and the entropy in this case is:

$$H\left(\frac{1}{D}, \frac{1}{D}, \ldots, \frac{1}{D}\right) = -D\left(\frac{1}{D}\log\frac{1}{D}\right) = \log D. \tag{34}$$

Thus, we have the inequality:

$$0 = H(1, 0, \ldots, 0) \leq H(\sigma_1, \sigma_2, \ldots, \sigma_D) \leq \log D. \tag{35}$$

The *effective rank* is defined as:

$$\text{erank}(\Sigma_{\mathbf{X}}) = \exp(H(\sigma_1, \sigma_2, \ldots, \sigma_D)), \tag{36}$$

which quantifies the "effective" number of singular values that are significantly contributing to the rank of the matrix. Since $H(\sigma_1, \sigma_2, \ldots, \sigma_D)$ is bounded by $\log D$, it follows that the effective rank is bounded by:

$$1 \leq \text{erank}(\Sigma_{\mathbf{X}}) \leq D. \tag{37}$$

Equality holds at the lower bound if and only if $(\sigma_1, \sigma_2, \ldots, \sigma_D) = (1, 0, \ldots, 0)$, that is, when all but one singular value is zero. In this case, the singular value vector is:

$$\sigma = (\|\sigma\|_1, 0, \ldots, 0)^T, \tag{38}$$

where $\|\sigma\|_1 = 1$. Hence, $\mathrm{erank}(\boldsymbol{\Sigma}_{\mathbf{X}}) = 1$.

Next, suppose that only $k$ singular values of $A$ are non-zero for some $k \in \{1, 2, \ldots, D\}$. In this case, the rank of $A$ is given by $\mathrm{rank}(A) = k$, and the entropy only depends on the non-zero singular values. Thus, we have:

$$H(\sigma_1, \sigma_2, \ldots, \sigma_D) = H(\sigma_1, \sigma_2, \ldots, \sigma_k), \tag{39}$$

where $\sigma_1, \sigma_2, \ldots, \sigma_k$ are the non-zero singular values. Since entropy is maximized when these non-zero singular values are uniformly distributed, we have:

$$H(\sigma_1, \sigma_2, \ldots, \sigma_k) \leq \log k. \tag{40}$$

Hence, the effective rank satisfies:

$$\mathrm{erank}(\boldsymbol{\Sigma}_{\mathbf{X}}) \leq \mathrm{rank}(\boldsymbol{\Sigma}_{\mathbf{X}}) \leq D, \tag{41}$$

with equality $\mathrm{erank}(\boldsymbol{\Sigma}_{\mathbf{X}}) = \mathrm{rank}(\boldsymbol{\Sigma}_{\mathbf{X}})$ if and only if the non-zero singular values are uniformly distributed, i.e.,

$$(\sigma_1, \ldots, \sigma_k, \sigma_{k+1}, \ldots, \sigma_D) = \left(\frac{1}{k}, \ldots, \frac{1}{k}, 0, \ldots, 0\right), \tag{42}$$

or equivalently:

$$\sigma = (\|\sigma\|_1/k, \ldots, \|\sigma\|_1/k, 0, \ldots, 0)^T. \tag{43}$$

In this case, the effective rank coincides with the actual rank of the matrix, since the singular values contribute equally to the rank.

# D   ABLATION STUDY

## D.1   NUMBER OF HEAD GROUPS

| Llama2-7B-chat-hf, KV size=384 | | | | | | | |
|---|---|---|---|---|---|---|---|
| **Group num** | **KV size in different groups** | **Qasper** | **HotpotQA** | **QMSum** | **SAMSum** | **Lcc** | **Average** |
| 2 groups | 32/736 | 18.23 | 30.96 | 19.82 | 40.05 | 57.13 | 33.24 |
| 3 groups | 32/384/736 | 18.90 | 30.53 | 19.95 | 40.04 | 58.29 | 33.54 |
| 4 groups | 32/266/502/736 | 19.29 | 30.48 | 20.10 | 41.03 | 59.37 | 34.05 |
| 5 groups | 32/208/384/560/736 | 19.58 | 31.17 | 20.72 | 40.61 | 58.93 | 34.20 |
| 8 groups | 32/132/232/332/436/536/636/736 | 19.34 | 31.04 | 20.16 | 40.92 | 59.48 | 34.19 |
| 2 groups | 256/512 | 19.67 | 30.98 | 20.20 | 39.36 | 60.28 | 34.10 |
| 3 groups | 256/384/512 | 19.45 | 31.29 | 20.24 | 39.63 | 59.63 | 34.05 |
| 4 groups | 256/342/427/512 | 19.20 | 30.99 | 20.10 | 39.33 | 59.71 | 33.87 |
| 5 groups | 256/320/384/448/512 | 19.71 | 30.95 | 20.22 | 39.60 | 59.99 | 34.09 |
| 8 groups | 256/296/332/368/404/440/476/512 | 19.55 | 31.02 | 20.59 | 39.10 | 59.39 | 33.93 |

Table 6: Multiple group comparison

In this section, we analyze the impact of the number of groups on performance. As illustrated in the Table 6, when the KV size is set to 384 and the difference between the maximum and minimum KV sizes within each group is minimal, the number of groups has a small effect on overall performance, with the maximum observed variation being only 0.23%. However, when there is a significant disparity between the maximum and minimum KV sizes, increasing the number of groups tends to enhance performance, with a maximum observed improvement of 0.96%. This indicates that the number of groups is highly correlated with the distribution of KV sizes within groups, impacting the experimental results.

## D.2   TRUNCATION STRATEGY

In this section, we examine truncation strategies, with a focus on evaluating the effectiveness of elbow points. We conduct tests using various elbow points by selecting different top k eigenvalues and compared the results to cases where no elbow points are applied. As demonstrated in Table 7, the results demonstrate a 0.70% performance gap between the truncated and untruncated settings, highlighting the efficacy of our approach.

| Llama-2-7B-chat-hf, KV size=384 | | | | |
|---|---|---|---|---|
| **Top k** | **Qasper** | **QMSum** | **SAMSum** | **Lcc** | **Average** |
| top 16 | 19.28 | 20.38 | 39.45 | 59.72 | 34.71 |
| top 32 | 19.34 | 20.51 | 39.35 | 59.93 | 34.78 |
| top 64 | 18.75 | 20.43 | 39.36 | 59.86 | 34.60 |
| top all | 18.14 | 20.14 | 38.52 | 59.51 | 34.08 |

Table 7: Truncation strategy

### D.3 RANDOM PARTITION

In this section, we evaluate the validity of our layer partitioning approach. For this analysis, Llama2-7B-chat-hf is selected as our base model, where the KV size per layer is configured to 64. the attention heads are divided into two distinct groups: one group with a KV size of 96 and another with a KV size of 32. As demonstrated in Table 8, the performance gains achieve through our method are substantial, highlighting the effectiveness of our partitioning approach.

|  | Ours-group | Random-group |
|---|---|---|
| Qasper | 15.40 | 11.70 |
| Musique | 6.96 | 4.47 |
| GovReport | 14.88 | 7.31 |
| TREC | 55.50 | 32.50 |
| PCount | 5.00 | 3.97 |
| Lcc | 50.00 | 32.50 |
| Avg. | 24.62 | 15.49 |

Table 8: Random groups condition

### D.4 COMPRESSION RATIO ALLOCATION BETWEEN HEAD GROUPS

In this section we discuss the allocation of compressibility between groups. Using the same experimental setup as the previous section, we only exchange the KV size between the two groups, and find that the text generation exhibits abnormal changes. As demonstrated in Table 9, the results in the table indicate that text generation exhibited abnormalities in several datasets, with the overall average accuracy decreasing to 5.89%. This suggests that assigning smaller KV sizes to lower-rank groups is effective. Conversely, allocating smaller KV sizes to higher-rank groups leads to significant information loss.

|  | Ours-group | Random-group |
|---|---|---|
| Qasper | 15.40 | 2.32 |
| Musique | 6.96 | 0.13 |
| GovReport | 14.88 | 1.01 |
| TREC | 55.50 | 19.00 |
| PCount | 5.00 | 0.33 |
| Lcc | 50.00 | 12.50 |
| Avg. | 24.62 | 5.89 |

Table 9: Compressibility distribution

### D.5 MATRIX ENTROPY AND VARIANCE

| Methods | Single-Document QA | | | Multi-Document QA | | | Summarization | | | Few-shot Learning | | | Synthetic | | Code | | Avg. |
|---|---|---|---|---|---|---|---|---|---|---|---|---|---|---|---|---|---|
| | NtrvQA | Qasper | MF-en | HotpotQA | 2WikiMQA | Musique | GovReport | QMSum | MultiNews | TREC | TriviaQA | SAMSum | PCount | PRe | Lcc | RB-P | |
| Variance KV (384) | 16.75 | 18.15 | 32.09 | **32.42** | 27.29 | 8.50 | 19.46 | 20.42 | 22.94 | 62.50 | **84.65** | 38.64 | **5.50** | **12.00** | 58.59 | 52.98 | 32.06 |
| Uncomp (384) | **17.33** | **19.34** | **34.16** | 31.54 | **28.23** | **10.04** | **20.38** | **20.51** | **23.33** | **63.00** | 84.11 | **39.35** | **5.50** | 9.50 | **59.93** | **54.87** | **32.57** |
| Variance KV (64) | 8.75 | 13.58 | 12.24 | 20.27 | 13.38 | 3.89 | 8.76 | 15.73 | 13.98 | 29.50 | 56.22 | 30.35 | **5.00** | 5.45 | 37.10 | 30.47 | 19.04 |
| Uncomp (64) | **14.05** | **15.40** | **25.56** | **26.28** | **21.96** | **6.96** | **14.88** | **18.83** | **17.58** | **55.50** | **81.61** | **34.74** | **5.00** | **5.00** | **50.00** | **45.55** | **27.43** |

Table 10: Comparison of entropy and variance of truncated matrices

In this section we discuss the grouping policy. We provide the compression rate estimates based on the variance of attention scores in Table 10, evaluated under two KV Cache sizes, 384 and 64. The results clearly highlight our advantages, especially under the budget of 64. This suggests that solely relying on compression rate estimation based on attention is unreasonable, as attention itself is subject to biases such as the attention sink(Xiao et al., 2023) and recency bias(Peysakhovich & Lerer, 2023). It is necessary to introduce additional metrics to measure unbiased compression estimation methods.

### D.6 ATTENTION SCORE MATRIX SELECTION FOR HIDDEN STATES

| Methods | Single-Document QA | | | Multi-Document QA | | | Summarization | | | Few-shot Learning | | | Synthetic | | Code | | Avg. |
|---|---|---|---|---|---|---|---|---|---|---|---|---|---|---|---|---|---|
| | NtrvQA | Qasper | MF-en | HotpotQA | 2WikiMQA | Musique | GovReport | QMSum | MultiNews | TREC | TriviaQA | SAMSum | PCount | PRe | Lcc | RB-P | |
| Last Layer Prediction | **17.61** | **20.39** | 33.56 | 30.52 | **26.75** | **9.91** | 20.42 | **20.55** | **23.54** | **63.00** | 82.51 | 38.16 | **4.50** | 8.00 | 59.76 | 52.55 | 31.98 |
| Current Layer Prediction | 17.48 | 18.25 | **34.17** | **32.47** | 26.19 | 8.94 | **20.46** | 20.23 | 23.26 | 61.00 | **84.03** | **39.58** | **4.50** | **10.00** | **60.00** | **54.23** | **32.17** |

Table 11: Using the attention of the current layer and the attention of the previous layer on Long-Bench

We designed an experiment to demonstrate that using the attention scores from the previous layer to predict the compression strategy for the current layer is reasonable, as shown in the Table11. The performance difference between the two methods is minimal, but using the current layer for prediction results in inference being twice as slow and more computationally expensive. Therefore, Our method remains efficient while achieving good performance.

# E  THROUGHPUT ANALYSIS

| Llama2-7B-chat-hf, KV Size = 384, Prompt+Generate is 3712+384 | | | | | | |
|---|---|---|---|---|---|---|
| batch_size | Ours-group-stage | | Ours-group | | FullKV | |
| | ms/token | max memory used(MB) | ms/token | max memory used(MB) | ms/token | max memory used(MB) |
| 1 | 28.055 | 23492 | 28.536 | 23080 | 25.691 | 24690 |
| 4 | 7.910 | 37526 | 8.504 | 37516 | 13.806 | 44220 |
| 8 | 4.822 | 59014 | 5.436 | 59036 | 11.823 | 72444 |
| 10 | 5.340 | 69802 | 5.887 | 69780 | - | Out-of-Memory |
| 12 | 3.994 | 80514 | 4.567 | 80522 | - | Out-of-Memory |
| Llama2-7B-chat-hf, KV Size = 384, Prompt+Generate is 4032+64 | | | | | | |
| batch_size | Ours-group-stage | | Ours-group | | FullKV | |
| | ms/token | max memory used(MB) | ms/token | max memory used(MB) | ms/token | max memory used(MB) |
| 1 | 34.782 | 24298 | 39.231 | 24312 | 36.596 | 24240 |
| 4 | 13.671 | 41180 | 18.458 | 41170 | 23.952 | 41146 |
| 8 | 10.186 | 66560 | 15.074 | 66580 | 21.944 | 66532 |
| 10 | 9.907 | 79198 | 14.603 | 79206 | 21.482 | 79168 |
| 12 | 9.464 | 79140 | 14.150 | 79174 | - | Out-of-Memory |

Table 12: Throughput analysis

To ensure the accuracy of performance analysis, we conduct experiments on a NVIDIA A100 80G GPU. We randomly sample 96 data points from the Wikitext-2 dataset, with strict control over the token lengths for both the prompt and generation phases. Detailed analyses of memory usage and throughput are provided in the Table 12 . From the table, we can see that in the long-prompt, short-generate scenario, our method achieves up to 2.96x throughput.

In addition to the configurations outlined in the table, we conduct experiments under the 2048+8096 setting too. The results demonstrate that our method supports a batch size of 32, whereas FullKV is limited to a batch size of 6. Notably, in this scenario, FullKV requires 15.67ms per token generation, while our approach reduces this to only 2.45ms. Our methods leads to a throughput that is up to 6.4 times that of FullKV.

In this experiment, for the Ours-group-stage method, $S_{min}$ was set to 512, and the layers were divided into five groups: layers 0 to 1, layers 2 to 14, layers 15 to 25, layers 26 to 30, and layer 31. The reserved sizes for these groups were set to 4096, 3200, 2304, 1408, and 512, respectively. The attention heads were divided into two groups: one group with a KV size of 512, and the other with a KV size of 256. The final KV cache length retained by each layer was 384.

# F  SUPPLEMENTARY DATASET COMPARISO

## F.1  RULER

RULER(Hsieh et al., 2024) is a novel synthetic benchmark designed to comprehensively evaluate the capabilities of long-context language models (LMs). Unlike the traditional Needle-in-a-Haystack (NIAH) test, which focuses solely on retrieval tasks, RULER provides flexible configurations to support customized sequence lengths and task complexities. It extends the vanilla NIAH test by introducing diverse variations in the types and quantities of "needles" and adding new task categories, such as multi-hop tracing and aggregation, to assess capabilities beyond simple context search. Results are showed at Table 13, where the Llama-3-8B-Instruct model is used, and other Settings are consistent with the previous section A.3. The experiments are implemented on a single A100 80G GPU.

| RULER(8k) | niah_single_1 | niah_single_2 | niah_single_3 | niah_multikey_1 | niah_multikey_2 | niah_multikey_3 | niah_multivalue | niah_multiquery | vt | cwe | fwe | qa_1 | qa_2 | average |
|---|---|---|---|---|---|---|---|---|---|---|---|---|---|---|
| FullKV 8k | 100.00 | 100.00 | 100.00 | 98.80 | 88.20 | 97.60 | 95.40 | 99.40 | 98.60 | 97.74 | 83.93 | 67.40 | 50.80 | 90.61 |
| uncomp | **100.00** | **99.80** | 3.80 | **99.40** | **72.80** | 0.00 | **81.55** | **74.75** | 93.88 | 20.78 | 53.93 | 64.40 | 49.60 | **62.67** |
| snapkv | **100.00** | **99.80** | 1.60 | 98.80 | 72.60 | 0.00 | 78.00 | 71.05 | **94.36** | 21.16 | 49.60 | **64.80** | **50.00** | 61.67 |
| pyramidkv | **100.00** | 98.40 | 0.00 | 98.40 | 66.00 | 0.00 | 63.60 | 42.55 | 81.96 | 8.16 | 41.00 | 65.00 | 48.60 | 54.90 |
| chai | 35.00 | 22.80 | **23.40** | 22.00 | 3.80 | 0.60 | 23.40 | 23.80 | 11.24 | 0.66 | 7.00 | 25.80 | 21.80 | 17.02 |
| h2o | 2.80 | 3.80 | 5.80 | 5.40 | 4.00 | **3.00** | 4.60 | 5.20 | 4.60 | **34.60** | **85.87** | 42.00 | 39.60 | 18.56 |

| RULER(4k) | niah_single_1 | niah_single_2 | niah_single_3 | niah_multikey_1 | niah_multikey_2 | niah_multikey_3 | niah_multivalue | niah_multiquery | vt | cwe | fwe | qa_1 | qa_2 | average |
|---|---|---|---|---|---|---|---|---|---|---|---|---|---|---|
| FullKV 4k | 100.00 | 100.00 | 100.00 | 99.40 | 100.00 | 98.80 | 99.15 | 99.85 | 99.72 | 99.80 | 94.20 | 81.40 | 58.00 | 94.64 |
| uncomp | 100.00 | **99.80** | 18.80 | 95.60 | **98.80** | 0.00 | **93.00** | **93.00** | **95.84** | **56.06** | **78.07** | **81.40** | **57.20** | **74.43** |
| snapkv | **100.00** | 99.60 | 8.00 | **99.40** | 97.40 | 0.00 | 88.30 | 87.70 | 95.80 | 52.86 | 76.33 | **81.40** | 56.60 | 72.57 |
| pyranikv | **100.00** | 99.40 | 0.60 | 98.60 | 91.80 | 0.00 | 65.85 | 49.40 | 78.84 | 10.50 | 66.20 | 81.00 | 55.40 | 61.35 |
| chai | 44.40 | 54.00 | **46.60** | 36.60 | 14.00 | **7.20** | 53.40 | 52.60 | 17.16 | 13.00 | 25.60 | 59.40 | 30.20 | 34.94 |
| h2o | 10.40 | 12.60 | 13.00 | 14.60 | 9.20 | 7.00 | 12.25 | 13.15 | 8.64 | 82.94 | **93.00** | **81.80** | 40.00 | 30.66 |

Table 13: Performance comparison of methods on RULER benchmark across different context lengths. The first section shows results for an 8k context, while the second section highlights 4k context performance.

## F.2 INFINITEBENCH

| Method | En.Sum | En.QA | En.MC | En.Dia | Zh.QA | Code.Debug | Code.Run | Math.Calc | Math.Find | Retrieve.PassKey | Retrieve.Number | Retrieve.KV | Average |
|---|---|---|---|---|---|---|---|---|---|---|---|---|---|
| FullKV | 12.55 | 0.27 | 42.79 | 1.00 | 4.04 | 22.34 | 0.00 | 0.00 | 38.57 | 6.27 | 6.44 | 4.80 | 14.38 |
| uncomp | **11.74** | 0.23 | **44.98** | 3.80 | 3.00 | 21.57 | 0.00 | 0.00 | **38.57** | **6.27** | 6.44 | 0.00 | **14.77** |
| snapkv | 11.59 | **0.28** | 42.36 | 1.00 | 4.01 | 21.83 | 0.00 | 0.00 | 38.29 | **6.27** | 6.61 | 0.00 | 14.22 |
| pyramidkv | 11.34 | 0.23 | 40.61 | 2.50 | **4.03** | 22.08 | 0.00 | 0.00 | **38.57** | **6.27** | **6.78** | 0.00 | 14.26 |
| chai | 9.69 | **0.37** | 34.06 | **8.00** | 3.26 | **24.97** | 0.00 | 0.00 | 27.43 | 4.58 | 5.93 | 1.20 | 12.79 |
| h2o | 10.99 | 0.18 | **44.98** | 3.50 | 3.98 | 22.08 | 0.00 | 0.00 | 37.71 | 1.69 | 1.69 | 0.00 | 14.24 |

Table 14: Performance comparison of various methods on InfiniteBench across different tasks, including summarization, QA, mathematical reasoning, and code-related benchmarks. The "Average" column represents the overall average performance.

InfiniteBench(Zhang et al., 2024b) is a state-of-the-art benchmark designed to evaluate language models' ability to process, understand, and reason over extremely long contexts exceeding 100k tokens. By pushing context lengths 10 times beyond traditional datasets, InfiniteBench aims to advance applications of LLMs and enable high-level interactions in scenarios requiring extensive context comprehension. Results are showed at Table 14, where the Llama-3-8B-Instruct model is used, and other Settings are consistent with the previous section A.3.

## G SUPPLEMENTARY METHOD COMPARISON

| Methods | Single-Document QA | | | Multi-Document QA | | | Summarization | | | Few-shot Learning | | | Synthetic | | Code | | Avg. |
|---|---|---|---|---|---|---|---|---|---|---|---|---|---|---|---|---|---|
| | NtrvQA | Qasper | MF-en | HotpotQA | 2WikiMQA | Musique | GovReport | QMSum | MultiNews | TREC | TriviaQA | SAMSum | PCount | PRe | Lcc | RB-P | |
| FullKV | 19.34 | 18.89 | 35.19 | 30.66 | 28.26 | 10.05 | 25.27 | 20.22 | 25.86 | 63.00 | 83.62 | 41.70 | 5.00 | 10.00 | 61.40 | 55.41 | 33.37 |
| StreamLLM(Xiao et al., 2023) | 13.71 | 13.68 | 19.40 | 26.97 | 28.03 | 6.78 | 15.13 | 18.87 | 18.27 | 46.50 | 80.02 | 40.85 | 4.50 | 5.00 | 56.84 | 51.56 | 27.88 |
| Double-Sparse+key(Yang et al., 2024) | 17.27 | 19.85 | 32.30 | 29.45 | 28.54 | 9.90 | 20.88 | 19.84 | 25.38 | 61.50 | 83.48 | 40.56 | 5.25 | 8.00 | 52.27 | 51.97 | 31.65 |
| Double-Sparse+query(Yang et al., 2024) | **17.99** | 19.27 | 31.93 | 30.83 | 27.91 | 9.25 | **23.68** | **20.54** | **26.10** | 62.00 | **84.60** | **41.75** | 4.75 | 8.00 | 58.27 | **55.49** | 32.65 |
| Quest(Tang et al., 2024) | 17.31 | **19.55** | 32.18 | 30.25 | 27.20 | 9.48 | 22.82 | 19.25 | 25.99 | 62.50 | 83.26 | 40.37 | 5.00 | 5.25 | 58.81 | 53.24 | 32.03 |
| UNComp+key | 17.04 | 19.11 | 34.03 | 30.73 | **28.73** | 9.61 | 20.38 | 20.34 | 23.48 | **63.00** | 84.14 | 38.63 | **5.50** | **10.00** | 59.89 | 53.93 | 32.41 |
| UNComp+query | 17.33 | 19.34 | **34.16** | **31.54** | 28.23 | **10.04** | 20.38 | 20.51 | 23.33 | **63.00** | 84.11 | 39.35 | **5.50** | 9.50 | **59.93** | 54.87 | 32.57 |

Table 15: The supplementary baseline is uniformly compared with a kv cache size of 384. The base model is Llama2-7B-chat-hf. "(Method) + key" uses the features of the key matrix for model parameter pruning, while "(method) + query" uses the features of the query matrix for model parameter pruning. We conducted experiments using the default hyperparameters from the open-source code repository.

Double-sparse and our method achieved the best performance when using query-based sparse pruning, surpassing our method by 0.08%. However, when using the key matrix as the criterion for sparse pruning, its performance was worse than ours.

# H  PYRAMIDKV WITH UNCOMP

| Methods | Single-Document QA | | | Multi-Document QA | | | Summarization | | | Few-shot Learning | | | Synthetic | | Code | | Avg. |
| | NtrvQA | Qasper | MF-en | HotpotQA | 2WikiMQA | Musique | GovReport | QMSum | MultiNews | TREC | TriviaQA | SAMSum | PCount | PRe | Lcc | RB-P | |
| --- | --- | --- | --- | --- | --- | --- | --- | --- | --- | --- | --- | --- | --- | --- | --- | --- | --- |
| PyramidKV+Uncomp Group32 heads | **18.21** | **18.59** | 34.22 | 30.66 | 28.20 | 9.10 | 20.04 | 20.19 | 22.84 | **63.00** | **84.39** | 39.46 | **5.50** | 5.00 | 58.89 | 52.86 | 31.95 |
| PyramidKV+Uncomp Group8 heads | 17.74 | 18.00 | **34.48** | 31.26 | **28.21** | 9.20 | **20.41** | 20.38 | **23.23** | **63.00** | 84.21 | **40.30** | **5.50** | 8.50 | **59.38** | **53.65** | **32.34** |
| PyramidKV+Uncomp Group3 heads | 17.52 | 18.24 | 33.78 | **31.60** | 27.50 | **9.21** | 20.03 | 20.23 | 22.97 | **63.00** | 83.98 | 39.10 | **5.50** | **10.50** | 59.16 | 53.18 | 32.22 |
| PyramidKV+Uncomp Group2 heads | 17.19 | 17.93 | 33.25 | 30.70 | 27.62 | 9.05 | 20.19 | **20.84** | 22.79 | **63.00** | 84.03 | 38.35 | **5.50** | 9.50 | 59.23 | 53.10 | 32.02 |
| PyramidKV | 16.86 | 18.26 | 31.01 | 31.59 | 27.93 | 8.69 | 19.88 | 20.15 | 22.43 | 62.00 | 83.86 | 38.98 | **5.50** | 10.00 | 58.94 | 52.80 | 31.81 |

Table 16: The impact of PyramidKV's dynamic sparsity ratio when applied in UNComp. The comparison of dynamic sparsity ratios between this method and PyramidKV.

Based on PyramidKV+Uncomp Group8 heads, which applies PyramidKV using our method of setting different compression rates for different heads, our method can bring greater improvements to PyramidKV if an appropriate number of groups is chosen. This is because different heads can be categorized as streaming heads and retrieval heads (Xiao et al., 2024). It is reasonable for retrieval heads to compress fewer tokens with their groups. Setting finer-grained groups for compression might harm the performance of retrieval heads.

# I  ANALYSIS ABOUT MATRIX ENTROPY OF HIDDEN STATES

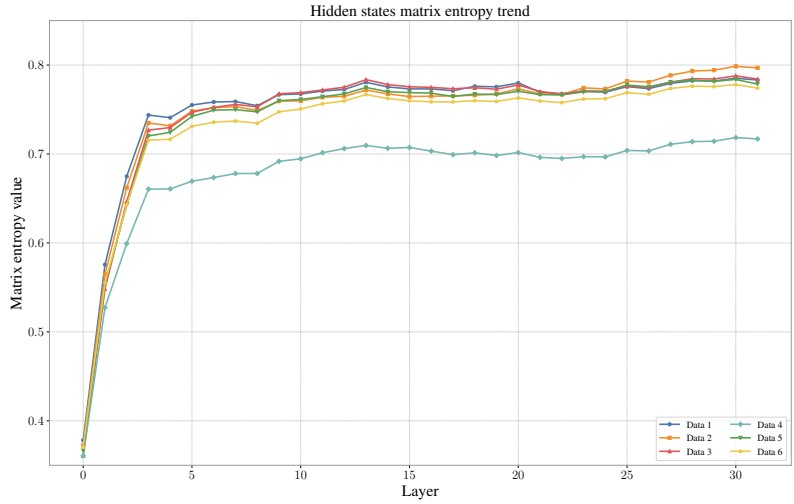

Figure 6: Matrix entropy of hidden states across different layers of Wikitext2 datasets

Figure 6 shows the matrix entropy trend of 6 samples of the Wikitext2 data set. It can be seen that the matrix entropy of hidden states increases layer by layer, which means that the token information becomes more and more abundant as the number of layers increases. This provides strong support to decrease the number of tokens retained by hidden states layer by layer.

