# OpenReview forum: "UNComp: Uncertainty-Aware Long-context Compressor for Efficient Large Language Model Inference"
_ICLR.cc/2025/Conference — Submitted to ICLR 2025_

### Official Review · Reviewer_PjLh · 2024-11-01

**Soundness:** 3
**Presentation:** 3
**Contribution:** 2
**Rating:** 6
**Confidence:** 5

**Summary:**

The paper finds that:
1. higher-layer tokens gather more information, and a small number of tokens can represent the entire sequence
2. For heads on the same layer, those with a higher effective rank should evict fewer tokens because this head is more informative
3. Tokens of the same head in different layers gradually share information as the layers deepen, while tokens of different heads do not share information as the layers deepen.

Therefore, based on the matrix entropy and effective rank, the KV cache and hidden states is compressed with training-free method, which achieves a compression rate of 4.74%, with a throughput increase of 6.4× and a 1.4× inference speedup in a single batch, incurring only a 1.41% performance loss.

**Strengths:**

The concept of matrix entropy and effective rank is novel and useful for determining the token redundancy.

**Weaknesses:**

The compression is based on the calculation of attention score and related accumulation, which introduces the onlince cost.

**Questions:**

The simplication of the calculation of importance score may be the fulture major direction.

---

### Official Review · Reviewer_xzeT · 2024-11-03

**Soundness:** 2
**Presentation:** 2
**Contribution:** 2
**Rating:** 5
**Confidence:** 4

**Summary:**

The paper presents UNComp, an innovative uncertainty-aware compression scheme designed to enhance the efficiency of large language models (LLMs) during long-context inference.

**Strengths:**

1. The writing is clear and easy to follow.
2. The source code is provided.

**Weaknesses:**

1. The number of groups seems to have little impact on performance, and sometimes fewer groups even yield better results. So why the complex design? However, if a uniform compression rate is applied, it feels like the paper doesn't contribute anything new.
2. Different layers have varying levels of attention to tokens, so "using the attention scores of the current layer to predict the tokens to be evicted in the next layer" may pose significant issues.
3. Lack of some baselines:  streamingLLM[1],Quest[2],doublesparse[3]

[1] Efficient Streaming Language Models with Attention Sinks https://arxiv.org/abs/2309.17453
[2] Quest: Query-Aware Sparsity for Efficient Long-Context LLM Inference https://arxiv.org/abs/2406.10774
[3] Post-Training Sparse Attention with Double Sparsity https://arxiv.org/abs/2408.07092

**Questions:**

1. This method has many hyperparameters; how did you select them?
2. If different heads retain a different number of tokens, does it affect parallel computation? If padding is used, how can true acceleration be achieved?
3. Why does a deeper layer necessarily retain fewer tokens? From the picture, it appears that the effective rank may fluctuate.

---

### Official Review · Reviewer_MMck · 2024-11-04

**Soundness:** 3
**Presentation:** 2
**Contribution:** 2
**Rating:** 6
**Confidence:** 3

**Summary:**

The paper introduces UNComp, an uncertainty-aware compression method designed to address memory and computational challenges associated with large language models (LLMs) during long-context inference. UNComp uses matrix entropy to estimate model uncertainty, applying selective compression across layers and attention heads based on these uncertainty levels. This approach preserves crucial information while enhancing efficiency in both memory and computational requirements.

**Strengths:**

- The paper introduces a matrix entropy to quantify the amount of information in each layer across the token sequence, which is then effectively applied for compression.
- Using the metric at both the layer and head levels, the authors propose customized inter-layer and inter-head compression strategies, allowing for a more targeted approach to model compression.
- The method undergoes extensive evaluation on diverse benchmarks, consistently delivering superior performance at comparable compression ratios.

**Weaknesses:**

* While Figure 3 aims to illustrate the overall workflow of the proposed method, it presents too much information at once, which makes it difficult to follow. One suggestion to improve readability is to break it down into subfigures or add step-by-step numbering to guide the reader through each part of the process. This adjustment would make the method’s workflow clearer and easier to understand.
* An essential aspect of evaluating compression methods is understanding the trade-off between accuracy and throughput (or latency). However, this paper separates these metrics: Table 1 presents only accuracy, while Table 3 focuses solely on latency, making it challenging to assess the accuracy-throughput balance across different methods at a glance. Adding a combined table or figure that displays both accuracy and throughput would better support comparisons of this trade-off.
* The paper primarily addresses end-to-end accuracy and latency but lacks an analysis of the compression ratio at each layer or head level within a single model (e.g., Llama3-8B-Instruct). Including this breakdown would provide greater insight into the internal dynamics and behavior of the model when applying the proposed method.
* Although the authors claim that the proposed method achieves faster performance than CHAI despite a lower compression ratio, the reasons for this improvement are not sufficiently explained. Offering more details on which specific aspects of the method contribute to greater hardware efficiency and speed, beyond just compression ratio, would make this claim more convincing.

**Questions:**

Please see the weaknesses section.

---

> ### Comment · Reviewer_MMck · 2024-11-27
>
> I have read the rebuttal and thanks for the correction and additional data and explanation. I will change the rating from 5 to 6.

---

### Official Review · Reviewer_NaPw · 2024-11-04

**Soundness:** 2
**Presentation:** 2
**Contribution:** 2
**Rating:** 5
**Confidence:** 4

**Summary:**

This paper focuses on the high latency and memory cost associated with long-context LLM inference by proposing UNComp, a training-free method that combines model compression and KV cache compression with a matrix entropy-based, dynamically allocated compression ratio. Specifically, the approach involves identifying similar layers and heads through offline search for compression, while using SnapKV with dynamic compression ratio of the KV cache with an approximated dynamic window size during inference. This paper test their method on the LongBench and NIAH benchmarks across four LLMs (Llama-2-7B/13B, Llama-3-8B, Mistral-7B-v0.1). Results indicate that UNComp offers slight improvements over baselines such as SnapKV, PyramidKV, and CHAI at the same compression ratio, although performance loss occurs when applying model compression.

**Strengths:**

- This paper focuses on a practical and relevant topic.
- The proposed matrix entropy-based method with dynamically allocated sparsity is intuitive.

**Weaknesses:**

1. The proposed approach is relatively straightforward and could be viewed as a combination of existing methods. For instance, the model compression component can function independently, yet no necessary baselines are provided for comparison in this area.
2. The paper lacks sufficient ablation studies and analysis to demonstrate the contribution of each module in the proposed method. Specifically:
  - The improvement over PyramidKV appears to mainly derive from the dynamic approximated window size selection based on matrix entropy. However, there is no ablation study examining the effect of applying dynamic approximated window sizes to PyramidKV, or the performance impact of applying PyramidKV’s dynamic sparse ratio within UNComp. A comparison of dynamic sparsity ratios between this method and PyramidKV is also missing.
  - There is no analysis of the dynamic approximated window size across different layers and heads.
3. The experiments are limited to LongBench and NIAH with a maximum context length of 32k, with no results for other state-of-the-art long-context benchmarks or longer contexts, such as RULER[1] and InfiniteBench[2].

[1] RULER: What’s the Real Context Size of Your Long-Context Language Models?
[2] InfiniteBench: Extending Long Context Evaluation Beyond 100K Tokens, ACL 2024.

**Questions:**

- **Q1**: Do you have any analysis of the dynamic sparsity ratio compared with PyramidKV?
- **Q2**: Do you have any analysis of the dynamic approximated window size across different layers and heads?
- **Q3**: Do you have results for other long-context benchmarks and longer context windows, such as RULER[1] and InfiniteBench[2]?
- **Q4**: Typo corrections needed for quotation marks, e.g., #390, #511, #713-715. And incorrect references, e.g., in Figure 3's legend, "Sec 4.2" might be "Sec 3.2" (#294, #298).

---

> ### Comment · Reviewer_NaPw · 2024-12-03
>
> Thank you for your response and the additional experiments provided. However, after carefully reviewing the supplementary experiments and considering comments from other reviewers, my concerns remain unresolved. The core insights of the paper are not clearly articulated, and there is insufficient evidence to justify the necessity and advantages of combining KV cache compression with model compression. Furthermore, the paper’s writing could be significantly improved to enhance its readability.
>
> In light of these considerations, I have decided to maintain my original score.

---

### Meta-Review · Area_Chair_afcg · 2024-12-20

**Metareview:**

The paper presents an innovative approach, UNComp, aimed at enhancing efficiency in large language models by introducing an uncertainty-aware compression scheme for long-context inference. This study contributes to the growing research on model efficiency in AI, addressing critical challenges in processing extensive token sequences.

The reviews indicated a mix of opinions, with some reviewers praising the clarity and potential applications of the proposed method, while others expressed concern over its novelty and practical implications. Notably, there was unanimous recognition of the paper's clear writing and provision of source code, but significant concerns remained about the complexity of the proposed design and its limited contributions compared to existing methods.

Despite the authors' efforts to address reviewer concerns in their rebuttal—such as clarifications on the entropy-guided framework and experimental results—certain critical issues remained unresolved. Reviewers continued to question the effectiveness of the proposed compression strategies and the justification for complexity that did not appear to lead to marked improvements over simpler models.
In conclusion, although the authors made commendable efforts in revising their manuscript and responding to feedback, significant concerns regarding the paper's novelty and practical contributions persist. Therefore, the recommendation is to reject the paper due to these unresolved issues.

**Additional Comments On Reviewer Discussion:**

The reviews indicated a mix of opinions, with some reviewers praising the clarity and potential applications of the proposed method, while others expressed concern over its novelty and practical implications. Notably, there was unanimous recognition of the paper's clear writing and provision of source code, but significant concerns remained about the complexity of the proposed design and its limited contributions compared to existing methods.

Despite the authors' efforts to address reviewer concerns in their rebuttal—such as clarifications on the entropy-guided framework and experimental results—certain critical issues remained unresolved.

While the authors expressed dissatisfaction with the negative reviewers and raised concerns about their professionalism, I believe that their misunderstanding of the content or evasion of responses did not reach a problematic level.

---

### Decision · Program_Chairs · 2025-01-22

Reject